# THINKING ON THE FLY: TEST-TIME REASONING ENHANCEMENT VIA LATENT THOUGHT POLICY OPTIMIZATION

**Wengao Ye**[1]*, **Yan Liang**[2], **Lianlei Shan**[3]
[1]University of Oxford, [2]University College London, [3]University of Chinese Academy of Sciences

## ABSTRACT

Recent advancements in Large Language Models (LLMs) have shifted from explicit Chain-of-Thought (CoT) reasoning to more efficient latent reasoning, where intermediate thoughts are represented as vectors rather than text. However, latent reasoning can be brittle on challenging, out-of-distribution tasks where robust reasoning is most critical. To overcome these limitations, we introduce Latent Thought Policy Optimization (LTPO), a parameter-free framework that enhances LLM reasoning entirely at test time, without requiring model parameter updates. LTPO treats intermediate latent "thought" vectors as dynamic parameters that are actively optimized for each problem instance. It employs an online policy gradient method guided by an intrinsic, confidence-based reward signal computed directly from the frozen LLM's own output distributions, eliminating the need for external supervision or expensive text generation during optimization. Extensive experiments on five reasoning benchmarks show that LTPO not only matches or surpasses strong baselines on standard tasks but also demonstrates remarkable robustness where others fail. Most notably, on highly challenging AIME benchmarks where existing latent reasoning baselines collapse to near-zero accuracy, LTPO delivers substantial improvements, showcasing a unique capability for complex reasoning.[1]

## 1 INTRODUCTION

Large Language Models (LLMs) have dramatically advanced the state of reasoning in artificial intelligence, propelled by **Chain-of-Thought (CoT)** prompting (Wei et al., 2022), which enhances performance by decomposing complex problems into explicit intermediate reasoning steps in natural language. Despite its effectiveness, the explicit generation of CoT is often inefficient and costly, both in computational resources and inference latency. To overcome this, recent research has focused on *latent reasoning*, which encodes the intermediate "thoughts" as continuous hidden vectors within the model's latent space. Notable approaches such as Coconut (Hao et al., 2024) and SoftCoT (Xu et al., 2025b) have demonstrated that by replacing textual CoT with learned latent embeddings, models can attain similar reasoning accuracy with increased computational efficiency.

However, a major limitation of existing latent reasoning approaches is their brittleness when handling challenging, out-of-distribution tasks. Methods relying on offline trained projection or latent modules typically fail to generalize to hard problems or new domains in practice, as shown in our experiments, advanced latent reasoning baselines can completely collapse on competition-grade benchmarks where robust reasoning is most critical.

In this work, we propose **Latent Thought Policy Optimization (LTPO)**, a parameter-free and entirely test-time framework for boosting LLM reasoning capabilities. Unlike prior latent reasoning approaches that fix the intermediate representations, LTPO treats the *latent thought vectors* as dynamic parameters to be actively optimized at test time. Concretely, given a prompt augmented with $K$ special *latent thought tokens*, we iteratively refine their hidden vectors in a closed-loop process: at each step, the current latent states are perturbed, passed through the frozen LLM (no model weight

---

*Correspondence to: Wengao Ye <wengao.ye@kellogg.ox.ac.uk>
[1]Source code: https://github.com/ltpo2025/LTPO

updates), and evaluated by an *intrinsic confidence-based reward* which is computed directly from the model's output distributions, without textual decoding. This reward signal guides a policy gradient update, improving the latent thoughts to maximize model's predictive certainty. After a small number of optimization steps, the refined latent thought vectors are concatenated with the prompt's embeddings, and passed through the LLM to generate the final answer.

LTPO is simple, general, and requires no additional data or model parameter updates, enabling *on-the-fly* enhancement of LLM reasoning abilities, purely via test-time latent space optimization. Extensive experiments on five mathematical reasoning benchmarks demonstrate that LTPO not only matches or surpasses strong baselines on standard tasks, but remains robust where others fail. Most notably, on highly challenging AIME competition benchmarks where existing latent reasoning methods collapse to near-zero accuracy, LTPO delivers substantial improvements: for instance, with Qwen-2.5-7B-Instruct (Yang et al., 2024), LTPO achieves 16.67% and 13.33% accuracy on AIME2024 and AIME2025 respectively, dramatically outperforming all competitive baselines.

Our key contributions are summarized as follows:

- We propose LTPO, a test-time framework for LLM reasoning, which directly optimizes intermediate latent thought vectors via policy gradient, without any model fine-tuning.
- We introduce an efficient online Reinforcement Learning (RL) loop using a self-derived confidence-based reward, computed from the LLM's own output distributions, eliminating the need for extra supervision or expensive text generation during optimization.
- We empirically show that LTPO delivers robust reasoning performance on diverse and challenging benchmarks, addressing critical failure points of previous latent reasoning approaches.

## 2 RELATED WORK

The pursuit of advanced reasoning in LLMs has evolved from generating explicit textual rationales, known as CoT prompting (Wei et al., 2022; Kojima et al., 2022; Zhou et al., 2023), to more efficient latent reasoning approaches. Latent reasoning represents intermediate "thoughts" as continuous vectors in the model's hidden space rather than as discrete text, aiming to match the performance of explicit CoT with lower latency. Most methods in this area rely on an offline training phase to learn these latent representations. For example, iCoT (Deng et al., 2024) fine-tunes a LLM on problems and their detailed reasoning steps to generate both the reasoning and final answer in a single forward pass; Coconut (Hao et al., 2024) directly trains the model to use the last hidden state of the "thought token" as the next input embedding; Similarly, SoftCoT (Xu et al., 2025b) utilizes a lightweight assistant model to generate soft thought tokens that are then projected into the LLM's latent space using a trained projection module. Although improving efficiency, these approaches can be brittle, especially on challenging, out-of-distribution problems where adaptability is most needed.

A parallel line of research has applied RL to enhance LLMs. The dominant paradigm, originating with Reinforcement Learning from Human Feedback (RLHF) (Ouyang et al., 2022; Bai et al., 2022), involves fine-tuning the model's parameters using algorithms like PPO (Schulman et al., 2017) to align with human preferences. This approach has been extended by more efficient methods like DPO (Rafailov et al., 2023) and specialized techniques for reasoning like GRPO (Shao et al., 2024; DeepSeek-AI et al., 2025) and DAPO (Yu et al., 2025). However, these methods universally operate by updating the model's weights in an offline training stage, based on external reward signals.

Our method, LTPO, diverges from both of these established research directions. It operates at the intersection of latent reasoning and optimization but introduces a new paradigm: enhancing reasoning entirely at test time without updating any model parameters. This approach is situated within the growing interest in test-time optimization strategies, which have recently been explored for instance-level adaption at test time (Li et al., 2025). However, LTPO operationalizes this concept through a distinct mechanism: it establishes a self-contained RL loop where the reward signal is derived directly from the LLM's own output distributions. This bypasses the need for explicit text generation or external evaluation within the RL loop, allowing for a more effective and efficient optimization of the latent reasoning path.

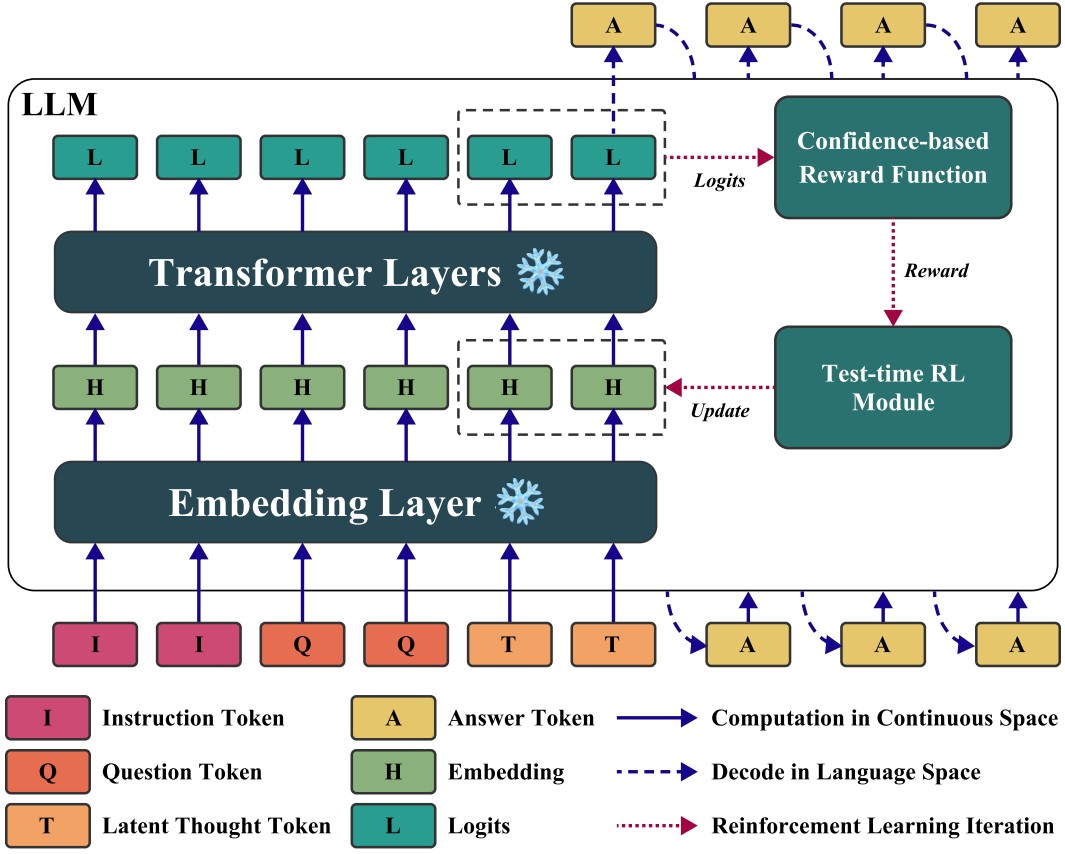

Figure 1: Overview of LTPO. The framework iteratively refines the embedding vectors of the *latent thought tokens* via a test-time RL loop. A confidence-based reward, calculated from the LLM's output logits, guides the Test-time RL Module to update the latent thought vectors. After optimization, the refined vectors are concatenated with the prompt's embeddings and passed through the LLM to generate the final answer.

## 3   METHOD

We introduce **LTPO**, a parameter-free framework that enhances the reasoning abilities of LLMs at test time. Figure 1 shows the overview of our framework.

### 3.1   LATENT THOUGHT REPRESENTATION

Let $\mathcal{M}_{\boldsymbol{\theta}}$ denote a LLM with frozen parameters $\boldsymbol{\theta}$. Given a standard input prompt which is a sequence of discrete tokens $\boldsymbol{x} = (x_1, ..., x_N)$, the model's embedding layer $E$ maps it to a sequence of vectors $E(\boldsymbol{x}) = (\boldsymbol{e}_1, ..., \boldsymbol{e}_N)$, where $\boldsymbol{e}_i \in \mathbb{R}^d$ and $d$ is the model's hidden dimension.

To enable latent reasoning, we augment the input prompt with $K$ special placeholder tokens, which we call *latent thought tokens*. Their initial embedding vectors, denoted by $\boldsymbol{H}^{(0)} = (\boldsymbol{h}_1^{(0)}, ..., \boldsymbol{h}_K^{(0)}) \in \mathbb{R}^{K \times d}$, are obtained by passing the placeholder tokens (e.g., [THINK]) through the model's embedding layer:

$$(\boldsymbol{h}_1^{(0)}, ..., \boldsymbol{h}_K^{(0)}) = E([\text{THINK}]_1, ..., [\text{THINK}]_K). \tag{1}$$

These latent thought vectors $\boldsymbol{H}$ serve as the parameters that will be optimized at test time. The full input to the LLM at any point is the concatenation of the original prompt embeddings and the current latent thought vectors, $E(\boldsymbol{x}) \| \boldsymbol{H}$.

### 3.2 OPTIMIZATION VIA TEST-TIME REINFORCEMENT LEARNING

We formulate the problem of finding the optimal latent thought vectors $\boldsymbol{H}$ as a sequential decision-making task, which we solve using a direct policy gradient method over $T$ timesteps.

**Policy as latent space exploration.** The optimization process iteratively refines the latent thought vectors $\boldsymbol{H}$, which themselves serve as the parameters of our policy. We define the components of this process as follows:

*State:* The state at timestep $t$ is the current set of latent thought vectors, $\boldsymbol{H}^{(t)} = (\boldsymbol{h}_1^{(t)}, ..., \boldsymbol{h}_K^{(t)})$.

*Action:* An action $\boldsymbol{A}^{(t)}$ is a candidate for the next state of the latent thoughts. The action space is continuous, where $\boldsymbol{A}^{(t)} = (\boldsymbol{a}_1^{(t)}, ..., \boldsymbol{a}_K^{(t)})$ and $\boldsymbol{a}_k^{(t)} \in \mathbb{R}^d$.

*Policy:* We employ a simple yet effective stochastic policy that explores the latent space by sampling actions from a multivariate Gaussian distribution centered at the current state $\boldsymbol{H}^{(t)}$:

$$\boldsymbol{A}^{(t)} \sim \pi(\cdot|\boldsymbol{H}^{(t)}) = \mathcal{N}(\boldsymbol{H}^{(t)}, \boldsymbol{\sigma}^2 \boldsymbol{I}), \tag{2}$$

where $\boldsymbol{\sigma}^2$ is a variance hyperparameter that controls exploration and is decayed over time. Sampling an action is equivalent to applying a random perturbation, $\boldsymbol{A}^{(t)} = \boldsymbol{H}^{(t)} + \boldsymbol{\epsilon}^{(t)}$, where $\boldsymbol{\epsilon}^{(t)} \sim \mathcal{N}(\boldsymbol{0}, \boldsymbol{\sigma}^2 \boldsymbol{I})$. This encourages local exploration around promising areas of the latent thought space.

**Confidence-based reward.** To guide the optimization without external supervision, we introduce an intrinsic reward signal derived directly from the LLM's own output distributions. Unlike recent works that leverage confidence for post-hoc selection (Kang et al., 2025; Fu et al., 2025), our application is distinct. LTPO employs confidence as a dense reward signal to actively guide an online optimization process within the latent space for each problem instance. The intuition is that a well-formed reasoning path should lead the model to be more confident in its subsequent predictions. While this intrinsic signal is powerful, we acknowledge that model confidence is not a perfect proxy for correctness, a potential limitation we discuss in Appendix B.

Given a candidate action $\boldsymbol{A}^{(t)}$ (a probe of next state), we compute the reward by first passing it through the frozen LLM $\mathcal{M}_{\boldsymbol{\theta}}$ along with the prompt embeddings $E(\boldsymbol{x})$ to obtain output probability distributions over the vocabulary $\boldsymbol{V}$ for each individual token:

$$(P_1^{(t)}, ..., P_{|\boldsymbol{x}|+K}^{(t)}) = \text{softmax}(\mathcal{M}_{\boldsymbol{\theta}}(E(\boldsymbol{x}) \parallel \boldsymbol{A}^{(t)})). \tag{3}$$

We define the confidence for a single latent thought vector $\boldsymbol{a}^{(t)}$ as the mean negative log-probability of the top-$k$ most probable tokens predicted from that position:

$$C(\boldsymbol{a}_i^{(t)}) = -\frac{1}{k} \sum_{v \in \text{top}(k, P_i^{(t)})} \log P_i^{(t)}(v), \tag{4}$$

where $\text{top}(k, P_i^{(t)})$ is the set of the $k$ most probable tokens in the distribution $P_i^{(t)}$. A higher confidence score indicates the model is more certain about its prediction. The final reward for the entire sequence of latent thoughts is the average confidence across all $K$ vectors:

$$R(\boldsymbol{A}^{(t)}) = \frac{1}{K} \sum_{i=1}^{K} C(\boldsymbol{a}_i^{(t)}). \tag{5}$$

**Latent thought update with policy gradient.** Since the reward function $R(\cdot)$ is non-differentiable with respect to the input latent vectors, we cannot use standard backpropagation. Instead, we can use a direct policy gradient method, such as REINFORCE (Williams, 1992), to update the policy parameters (i.e., the latent vectors). The objective is to find latent thought vectors $\boldsymbol{H}$ that maximize the expected reward:

$$J(\boldsymbol{H}) = \mathbb{E}_{\boldsymbol{A} \sim \pi(\cdot|\boldsymbol{H})}[R(\boldsymbol{A})]. \tag{6}$$

The gradient of this objective is given by the policy gradient theorem:

$$\nabla_{\boldsymbol{H}} J(\boldsymbol{H}) = \mathbb{E}_{\boldsymbol{A} \sim \pi(\cdot | \boldsymbol{H})}[R(\boldsymbol{A})\nabla_{\boldsymbol{H}} \log \pi(\boldsymbol{A} | \boldsymbol{H})]. \tag{7}$$

For our Gaussian policy (Equation 2), the log-probability gradient term can be simplified to:

$$\nabla_{\boldsymbol{H}} \log \pi(\boldsymbol{A} | \boldsymbol{H}) = \frac{(\boldsymbol{H} + \boldsymbol{\epsilon}) - \boldsymbol{H}}{\boldsymbol{\sigma}^2} = \frac{\boldsymbol{\epsilon}}{\boldsymbol{\sigma}^2}. \tag{8}$$

This yields a Monte Carlo estimate of the gradient from a single sample $\boldsymbol{\epsilon}^{(t)}$ at each optimization step $t$:

$$\nabla_{\boldsymbol{H}} J(\boldsymbol{H}^{(t)}) \approx R(\boldsymbol{H}^{(t)} + \boldsymbol{\epsilon}^{(t)})\frac{\boldsymbol{\epsilon}^{(t)}}{\boldsymbol{\sigma}^2}. \tag{9}$$

The latent thought vectors are then updated via standard gradient ascent:

$$\boldsymbol{H}^{(t+1)} = \boldsymbol{H}^{(t)} + \eta \cdot R(\boldsymbol{H}^{(t)} + \boldsymbol{\epsilon}^{(t)})\frac{\boldsymbol{\epsilon}^{(t)}}{\boldsymbol{\sigma}^2}, \tag{10}$$

where $\eta$ is the learning rate. This update rule intuitively pushes the latent thoughts $\boldsymbol{H}$ in the direction that yields higher rewards.

After $T$ optimization steps, the set of the optimized latent thought vectors $\boldsymbol{H}^*$ is concatenated with the prompt embeddings and passed through the LLM to autoregressively generate the final answer:

$$\boldsymbol{y} = \text{Decoder}(\mathcal{M}_{\boldsymbol{\theta}}(E(\boldsymbol{x}) \| \boldsymbol{H}^*)). \tag{11}$$

## 4 EXPERIMENTS

In this section, we present our experimental setup, results, and analysis to validate the effectiveness of LTPO. Addtional implementation details are provided in Appendix A.

### 4.1 EXPERIMENTAL SETUP

**Models.** We evaluate our method on four open-source reasoning LLMs from two distinct model families: LLaMA-3.1-8B-Instruct, LLaMA-3.2-3B-Instruct (Grattafiori et al., 2024), Qwen-2.5-7B-Instruct (Yang et al., 2024), and Qwen-3-14B (Yang et al., 2025). These models are fully open-source for reproducibility, covering multiple parameter scales to test robustness and generalizability.

**Datasets.** Following Deng et al. (2024) and Liu et al. (2025), we focus on mathematical reasoning for evaluation. We conduct experiments on five datasets: GSM8K (Cobbe et al., 2021), MATH-500 (Hendrycks et al., 2021), ASDiv (Miao et al., 2020), AIME2024 (Mathematical Association of America, 2024), and AIME2025 (Mathematical Association of America, 2025). GSM8K and MATH-500 are commonly used for evaluating the performance of reasoning systems, thus providing a solid basis for comparative analysis. ASDiv is a benchmark consisting of varied, multi-step elementary school math word problems. AIME2024 and AIME2025 are high-difficulty mathematical competition problems which are widely adopted in recent evaluations of top reasoning LLMs (e.g., Gemini-2.5 (Google, 2025), Grok-4 (xAI, 2025), GPT-5 (OpenAI, 2025)) and featured in the MathArena leaderboard (Balunović et al., 2025). Specifically, we utilize the augmented version of ASDiv (ASDiv-Aug) from Xu et al. (2025b) to ensure that the models encounter novel instances, given that LLaMA and Qwen are well-trained LLMs.

**Baselines.** We compare LTPO against three strong and directly comparable baselines to provide a thorough evaluation of its effectiveness: (1) **Zero-Shot CoT**: Our foundational baseline is the zero-shot CoT prompting method from Kojima et al. (2022). This approach represents the standard for eliciting reasoning in the *discrete token space* by instructing the model to generate explicit, step-by-step thinking. On modern LLMs, this technique sets a very strong performance baseline, often exceeding fine-tuned models, making it an essential point of comparison. (2) **Zero-Shot CoT-Unk**: We directly append untuned [UNK] tokens to the prompt to perform CoT reasoning. This

Table 1: Performance of LTPO vs. baselines across four models and five reasoning benchmarks, reported in accuracy (%). The optimal results are in bold and the suboptimal ones are underlined.

| Model | Method | GSM8K | MATH 500 | ASDiv Aug | AIME 2024 | AIME 2025 | Avg. |
|-------|--------|-------|----------|-----------|-----------|-----------|------|
| LLaMA-3.1-8B-Instruct | Zero-Shot CoT | 76.88 | 48.60 | 79.58 | 6.67 | 0.00 | 42.35 |
| | Zero-Shot CoT-Unk | 80.67 | 46.20 | 89.50 | 10.00 | 3.33 | 45.94 |
| | SoftCoT | 80.36 | 39.80 | 87.57 | 0.00 | 0.00 | 41.55 |
| | LatentSeek (Prompt 1) | 53.15 | **52.20** | 46.15 | 3.33 | 0.00 | 30.97 |
| | LatentSeek (Prompt 2) | **83.70** | 47.40 | **90.66** | 3.33 | 0.00 | 45.02 |
| | **LTPO (Ours)** | 81.27 | 49.00 | 89.69 | **16.67** | **6.67** | **48.66** |
| LLaMA-3.2-3B-Instruct | Zero-Shot CoT | 69.60 | 45.60 | 85.84 | 6.67 | 0.00 | 41.54 |
| | Zero-Shot CoT-Unk | 76.65 | 45.00 | 87.82 | 0.00 | 3.33 | 42.56 |
| | SoftCoT | 71.49 | 35.80 | 85.74 | 0.00 | 0.00 | 38.61 |
| | **LTPO (Ours)** | **76.88** | **46.40** | **88.63** | **13.33** | **6.67** | **46.38** |
| Qwen-2.5-7B-Instruct | Zero-Shot CoT | 88.55 | 72.40 | 91.71 | 10.00 | 10.00 | 54.53 |
| | Zero-Shot CoT-Unk | 86.20 | 70.80 | 91.52 | 13.33 | 10.00 | 54.37 |
| | SoftCoT | 85.29 | 65.20 | 88.92 | 0.00 | 0.00 | 47.88 |
| | LatentSeek (Prompt 1) | **89.54** | **73.60** | 91.43 | 10.00 | 0.00 | 52.91 |
| | LatentSeek (Prompt 2) | 86.20 | 57.80 | 91.81 | 6.67 | 6.67 | 49.83 |
| | **LTPO (Ours)** | 88.17 | **73.60** | 92.20 | **16.67** | **13.33** | **56.79** |
| Qwen-3-14B | Zero-Shot CoT | 91.96 | 68.20 | 92.77 | **10.00** | **10.00** | 54.59 |
| | Zero-Shot CoT-Unk | 91.66 | 71.20 | 94.03 | 6.67 | 0.00 | 52.71 |
| | SoftCoT | **93.78** | 72.60 | 91.62 | 0.00 | 0.00 | 51.60 |
| | **LTPO (Ours)** | 91.89 | **73.00** | **94.41** | **10.00** | **10.00** | **55.86** |

baseline isolates the contribution of our test-time optimization procedure for latent thought tokens. (3) **SoftCoT**: To provide a more direct and advanced comparison for our latent reasoning framework, we adopt SoftCoT (Xu et al., 2025b), which performs reasoning in the *continuous latent space*. It has been shown to outperform methods like LoRA fine-tuning (Hu et al., 2022) and Coconut (Hao et al., 2024) on various datasets. (4) **LatentSeek**: We also compare against LatentSeek (Li et al., 2025), which applies instance-level policy gradient optimization in the latent space at test time. Unlike our approach, LatentSeek relies on full autoregressive decoding to evaluate each intermediate step. Including this baseline allows us to highlight the critical efficiency and robustness advantages of our intrinsic confidence-based reward over computationally intensive decoding-based optimization methods.

**Evaluation.** All results are reported in accuracy (%). For each problem, we perform greedy decoding to generate the final answer. The generated output is then parsed to extract the final numerical answer, which is compared against the ground truth.

## 4.2 MAIN RESULTS

The comprehensive experimental results are presented in Table 1. Our proposed method, LTPO, demonstrates superior performance across all tested models and benchmarks. A key observation is that LTPO achieves the highest average accuracy in every model category, establishing its effectiveness and generalizability. For instance, with Qwen-2.5-7B-Instruct, LTPO achieves an average accuracy of 56.79%, surpassing Zero-Shot CoT by 2.26 points, SoftCoT by 8.91 points, and LatentSeek by 3.88 points. Similarly, with LLaMA-3.1-8B-Instruct, LTPO scores 48.66% on average, outperforming Zero-Shot CoT by 6.31 points, SoftCoT by 7.11 points, and LatentSeek by 3.64 points. These results strongly support that LTPO is an effective strategy for enhancing LLM reasoning. A deeper analysis of the results reveals several key strengths of the LTPO framework:

**Exceptional robustness on challenging problems.** The most notable result is LTPO's performance on the highly difficult AIME2024 and AIME2025 benchmarks. As hypothesized, these problems highlight the brittleness of existing latent reasoning methods. The advanced SoftCoT baseline completely collapses on these tasks, scoring 0.00% on AIME2024 and AIME2025 for all

Table 2: Performance of LTPO vs. training-based baselines on GSM8K, MATH-500, and AIME-2024 with Llama-3.1-8B-Instruct, reported in accuracy (%). The "Train Model Params" column indicates whether the method requires updating model parameters. The symbol † indicates the accuracy is reported by Zeng et al. (2025). The optimal results are in bold and the suboptimal ones are underlined.

| Method | Train Model Params | Supervision Type | GSM8K | MATH 500 | AIME 2024 |
|---|---|---|---|---|---|
| Genius | Yes | Self | 78.09 | 47.60 | 3.33 |
| SimpleRL-Zoo† | Yes | External Data | 79.20 | 23.00 | 0.00 |
| iCoT | Yes | External Data | 50.05 | - | - |
| SoftCoT | Yes | External Data | 80.36 | 39.80 | 0.00 |
| **LTPO (Ours)** | No | Self | **81.27** | **49.00** | **16.67** |

tested models. This result likely stems from the fact that SoftCoT's static projection module, which is trained on GSM8K data, does not generalize to the new domain of AIME competition mathematics. This highlights a key failure mode for trained latent reasoning methods when faced with out-of-distribution problems. While LatentSeek demonstrates competitive performance on simpler tasks like GSM8K, it also struggles to generalize to these competition-grade problems. In stark contrast, LTPO demonstrates remarkable resilience. With Qwen-2.5-7B-Instruct, LTPO achieves 16.67% on AIME2024 and 13.33% on AIME2025, dramatically outperforming not only the failed SoftCoT baseline but also the strong Zero-Shot CoT (10.00% on both). This unique capability to make substantial gains on problems where other methods fail underscores the value of our on-the-fly optimization process for navigating complex reasoning spaces. To further investigate the scalability of LTPO under a high-compute regime, we evaluate Qwen-3-14B on AIME benchmarks with the maximum generation length extended from 4,096 to 64,000 tokens in Section 4.7.

**Effectiveness of test-time optimization.** By comparing LTPO to the Zero-Shot CoT-Unk baseline, we can isolate the contribution of our optimization procedure. The Zero-Shot CoT-Unk method, which uses untuned latent thought tokens, underperforms LTPO for nearly all cases. For example, on LLaMA-3.2-3B-Instruct, LTPO's average accuracy of 46.38% is 3.82 points higher than Zero-Shot CoT-Unk's 42.56%. Similarly, on Qwen-3-14B, LTPO's average accuracy is 3.15 points higher than Zero-Shot CoT-Unk's 52.71%. This margin confirms that the performance gains are not merely from adding placeholder tokens but are fundamentally driven by the optimization of the latent thought vectors at test time.

**Generalizability across models.** The consistent superiority of LTPO across both LLaMA and Qwen model families, and at scales from 3B to 14B parameters, highlights the generalizability of our approach. LTPO does not rely on any model specific architecture or pretraining objective. Instead, it leverages the model's intrinsic confidence signal, a universal property of probabilistic models, making it broadly applicable across LLMs.

### 4.3 COMPARISON WITH TRAINING-BASED METHODS

To further situate LTPO within the landscape of reasoning methods, we conduct a targeted comparison against different paradigms on the LLaMA-3.1-8B-Instruct model. We introduce three additional baselines that require a model training phase and/or external training data: (1) **Genius** (Xu et al., 2025a) fine-tunes the model using a *self-reward* mechanism derived from the model's own reasoning process. (2) **SimpleRL-Zoo** (Zeng et al., 2025) fine-tunes the model using a *verifiable-reward* mechanism derived from external data. (3) **iCoT** (Deng et al., 2024) fine-tunes the model to internalize CoT steps for generating the reasoing and final answer in a single forward pass.

As shown in Table 2, LTPO outperforms all baselines on Llama-3.1-8B-Instruct. Notably, LTPO achieves these results without any updates to the model's parameters, operating entirely at test time. This stands in stark contrast to RL-based methods like Genius and SimpleRL-Zoo, which require a costly offline training phase. On GSM8K, LTPO's score of 81.27% surpasses both Genius (78.09%) and SimpleRL-Zoo (79.20%). While LTPO (49.00%) also leads on MATH-500, outperforming

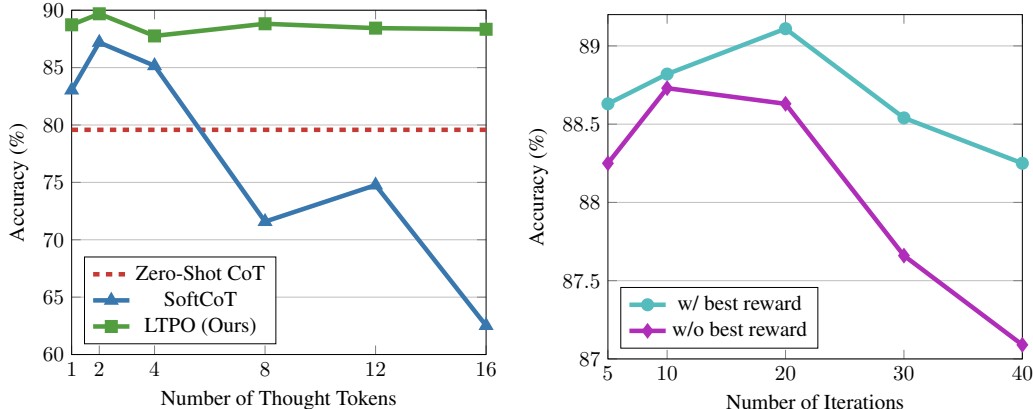

Figure 2: Left: The impact of thought token numbers. Right: The impact of LTPO using thought tokens with best reward. Both are tested on ASDiv-Aug using LLaMA-3.1-8B-Instruct.

Genius (47.60%) and more than doubling the score of SimpleRL-Zoo (23.00%), its advantage is even more stark on the highly challenging AIME2024 benchmark. Here, LTPO achieves 16.67% accuracy, dramatically outperforming both Genius (3.33%) and SimpleRL-Zoo (0.00%). These results demonstrate that our test-time optimization of latent thoughts offers a more effective and robust alternative to offline policy alignment via RL.

## 4.4 Analysis of Inference Efficiency

While test-time optimization can introduce inference overhead, LTPO is designed for efficiency. To quantify this, we benchmark its average inference time against baselines on GSM8K, AIME2024, and AIME2025 using the LLaMA-3.1-8B-Instruct model. We set the number of optimization steps to $T = 20$ for LTPO, a value that yields strong accuracy (as shown in Figure 2, right).

The results in Table 3 show that: (1) on the simpler GSM8K benchmark, LTPO, SoftCoT, and Zero-Shot CoT have nearly identical inference times (about 5.7–5.9 seconds), indicating the overhead of the LTPO loop is not substantial. In stark contrast, LatentSeek requires significantly more time (48.99–61.83 seconds), which is an order of magnitude slower. (2) on the challenging AIME benchmarks, which require much longer reasoning chains, the explicit Zero-Shot CoT method is slow, yet LatentSeek is prohibitively expensive, taking up to 1000 seconds per problem (LatentSeek with prompt 2 on AIME2024). Crucially, LTPO is not only faster than Zero-Shot CoT

Table 3: Average inference time per problem on GSM8K, AIME2024, and AIME2025 with LLaMA-3.1-8B-Instruct, reported in seconds. We set the RL steps $T = 20$ for LTPO, and $T = 10$ for LatentSeek to follow the setting in the original paper. Results are averaged on 3 runs per benchmark per method. The optimal results are in bold and the suboptimal ones are underlined. The "P1" and "P2" represent "Prompt 1" and "Prompt 2", respectively.

| Method | GSM8K | AIME 2024 | AIME 2025 | Avg. |
|---|---|---|---|---|
| Zero-Shot CoT | 5.81 | 62.59 | 55.60 | 41.33 |
| SoftCoT | 5.91 | **26.01** | 35.03 | 22.32 |
| LatentSeek (P1) | 48.99 | 872.47 | 708.40 | 543.29 |
| LatentSeek (P2) | 61.83 | 1002.30 | 512.43 | 525.52 |
| LTPO ($T = 20$) | **5.69** | 31.80 | **24.89** | **20.79** |

but is also computationally competitive with SoftCoT, achieving the lowest average inference time overall (20.79 seconds vs. over 500 seconds for LatentSeek).

LTPO's efficiency stems from a crucial difference in how computation is used: (1) Standard CoT methods rely on autoregressive decoding to generate a lengthy textual chain of thought. (2) LatentSeek relies on full text generation at each intermediate step during the optimization, leading to excessive latency. In contrast, the $T$ forward passes within the LTPO optimization loop do not involve any autoregressive decoding steps; instead, they are computationally efficient passes on a

fixed-length input (prompt + latent tokens) to calculate the reward, completely bypassing text generation within the optimization loop. (3) SoftCoT requires an assistant model to first generate soft thought tokens before the main model can process the problem and produce an answer, while LTPO operates within a single model, avoiding the multi-stage overhead of text generation. The final answer in the LTPO framework is decoded only once after the latent thought vectors have been optimized. Consequently, LTPO's entire test-time procedure is less expensive than Zero-Shot CoT and SoftCoT, and over 25 times faster than LatentSeek on average, demonstrating that it enhances reasoning without introducing a substantial latency penalty. We provide a further analysis of optimization overhead and generation savings between Zero-Shot CoT and LTPO in Appendix D.

## 4.5 ANALYSIS OF KEY COMPONENTS

We conduct further experiments to analyze the behavior of LTPO and its key components. These studies are performed on the ASDiv-Aug benchmark.

### 4.5.1 IMPACT OF THE NUMBER OF THOUGHT TOKENS

Both LTPO and SoftCoT use special tokens to represent reasoning in latent space. In Figure 2 (left), we investigate the effect of varying the number of these tokens. We observe that LTPO's performance is stable and robust, remaining high as the number of thought tokens increases from 1 to 16. In contrast, SoftCoT's performance degrades sharply when more than 4 thought tokens are used. This suggests that SoftCoT's static linear projection mechanism struggles to produce high-quality representations for a larger number of thought tokens, whereas LTPO's dynamic test-time optimization can effectively refine them toward an optimal state regardless of their quantity. We provide a further analysis of this stability in Appendix E.

### 4.5.2 IMPACT OF BEST REWARD SELECTION

LTPO iteratively updates the latent thought vectors to maximize a confidence-based reward. A key choice is whether to use the vectors from the final optimization step or the vectors that achieved the highest reward at any point during the process. Figure 2 (right) shows that using the thought tokens with the best reward consistently outperforms using those from the final iteration. This indicates that the optimization trajectory may not be monotonic, and the final state may not be the optimal one found during exploration. By tracking the highest confidence discovered during exploration, LTPO can more reliably elicit the best possible response from the model, a finding that aligns with similar observations by Kang et al. (2025). Furthermore, the diminishing returns from extended optimization steps point to a potential divergence between the model's predictive confidence and its actual reasoning correctness, a limitation we explore in Appendix B.

### 4.5.3 SENSITIVITY TO THE TOP-K REWARD HYPERPARAMETER

A critical component of LTPO is the confidence-based reward function (Equation 4), which relies on a single hyperparameter $k$. We analyze the sensitivity of LTPO to this top-$k$ hyperparameter by varying $k$ from 5 to 100 on ASDiv-Aug, with the results presented in Table 4.

Table 4: The impact of the top-$k$ hyperparameter on LTPO performance, reported in accuracy (%) on the ASDiv-Aug benchmark.

| Model | $k = 5$ | $k = 10$ | $k = 20$ | $k = 50$ | $k = 75$ | $k = 100$ |
|---|---|---|---|---|---|---|
| LLaMA-3.1-8B-Instruct | 88.92 | **89.11** | 88.63 | 88.92 | 88.73 | 88.63 |
| LLaMA-3.2-3B-Instruct | **87.48** | 87.28 | 87.19 | 87.09 | 87.38 | 86.99 |
| Qwen-2.5-7B-Instruct | **92.20** | 91.81 | 91.33 | 91.62 | 92.00 | 91.91 |

**High stability and robustness.** The primary observation is the remarkable stability of LTPO's performance across a wide range of $k$ values. For LLaMA-3.1-8B-Instruct, accuracy varies by less than 0.5 percentage points (from 88.63% to 89.11%) as $k$ increases from 5 to 100. This pattern holds for all tested models, indicating that the optimization process is not sensitive to the precise value of this hyperparameter.

**Optimal $k$ is in the small-to-medium range.**    While performance is generally stable, the optimal value for $k$ consistently appears in the smaller range ($k \leq 10$) across all models. This empirically validates that the most potent signal of model confidence is captured by the very highest probability tokens.

**Practical implications.**    This robustness is a practical advantage of LTPO. It reduces the overhead of hyperparameter tuning and suggests that the core benefit of the method comes from the directional signal of the confidence-based reward rather than its precise formulation. The optimization process appears to successfully capture the gradient of model certainty, whether that certainty is measured over the top 5 or top 100 tokens. This indicates that for a confident model, the probability mass is sufficiently concentrated in the first few tokens, and including more tokens from the long, low-probability tail of the distribution does not introduce significant noise to the reward signal.

### 4.6    GENERALIZATION TO OTHER DOMAINS

To verify that LTPO generalizes beyond mathematical domains, we evaluate it on StrategyQA (Geva et al., 2021) for commonsense reasoning and Date Understanding (Srivastava et al., 2023) from BIG-benchmark for symbolic reasoning.

As detailed in Appendix C, LTPO consistently outperforms the Zero-Shot CoT baseline across both tasks and models. Notably, on the symbolic reasoning task, LTPO with Qwen-2.5-7B-Instruct achieves 76.40% accuracy, delivering a substantial +12.0% improvement over the baseline. Crucially, the results highlight the adaptability of test-time optimization compared to fixed latent representations. While the training-based SoftCoT baseline degrades performance on StrategyQA, LTPO successfully improves upon the baseline. This confirms that LTPO is a robust, domain-agnostic framework capable of enhancing reasoning on tasks significantly distinct from standard mathematical benchmarks.

### 4.7    SCALABILITY WITH EXTENDED GENERATION LENGTH

We further investigate if LTPO remains beneficial when the base model is allowed ample compute for extensive reasoning. We evaluate Qwen-3-14B on AIME benchmarks with the maximum generation length extended to 64k tokens.

Table 5: Performance of LTPO vs. Zero-Shot CoT on AIME2024 and AIME2025 with Qwen-3-14B, reported in accuracy (%). The maximum output tokens is set to 64,000. The optimal results are in bold.

| Method | AIME 2024 | AIME 2025 | Avg. |
|---|---|---|---|
| Zero-Shot CoT | 80.00 | 70.00 | 75.00 |
| **LTPO (Ours)** | **83.33** | **76.67** | **80.00** |

As shown in Table 5, under this high-compute regime, the Zero-Shot CoT baseline improves dramatically to 75.00% average accuracy (a massive increase from the 10.00% accuracy observed under standard settings in Table 1). However, LTPO continues to provide significant gains, boosting performance to 80.00% (+5.0%) on average. This result suggests that LTPO is complementary to long-context explicit reasoning: even when the model can "think" extensively in natural language, optimizing the initial latent thought vectors provides a better starting point for the reasoning trajectory.

## 5    CONCLUSION

We introduce LTPO, a parameter-free framework that enhances LLM reasoning entirely at test time by directly optimizing latent thought vectors. LTPO employs an online policy gradient method guided by an intrinsic, confidence-based reward signal to refine its reasoning path for each problem without requiring model parameter updates. This approach overcomes the critical brittleness of existing latent reasoning methods, demonstrating robust reasoning ability on the highly challenging AIME benchmarks where LTPO delivers substantial accuracy gains while advanced latent reasoning baselines collapse to near-zero. Our work thus establishes test-time latent space optimization as a powerful and practical paradigm for eliciting more robust and effective reasoning from LLMs.

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

# A    EXPERIMENTAL DETAILS

This section provides a comprehensive overview of the experimental setup used to evaluate our proposed method, LTPO, ensuring full reproducibility of our results.

## A.1    COMPUTATIONAL RESOURCES

All experiments are conducted on a single A100-80GB or a single RTX4090-48GB GPU. All baseline models, including SoftCoT and iCoT, are trained on a single A100-80GB GPU. For Genius baseline, we directly use the official fine-tuned model [2] provided by Xu et al. (2025a).

## A.2    EVALUATION PROTOCOL

Performance is measured by accuracy (%). For each problem, we use greedy decoding to generate the response and set the maximum output length to 4096 tokens. A parser is used to extract the final numerical answer from the generated text, which is then compared against the ground-truth. A fixed random seed is used across all experiments to ensure consistency.

## A.3    BASELINE IMPLEMENTATION DETAILS

**Zero-Shot CoT.**    We follow the standard methodology proposed by Kojima et al. (2022). We append the same instruction to every problem prompt: "Please reason step by step, and put your final answer within \boxed{}."

**Zero-Shot CoT-Unk.**    This baseline uses the exact same prompt template as LTPO (shown in Appendix A.4), including the special placeholder tokens for latent thoughts. However, the embeddings for these tokens are not optimized at test time. This setup is designed to isolate the performance gains attributable specifically to our test-time RL optimization loop.

**SoftCoT.**    We use the official code[3] provided by Xu et al. (2025b). For benchmarks that lack a dedicated training set (MATH-500, AIME2024, and AIME2025), we follow the original authors' protocol and use the projection module trained on the GSM8K dataset.

**LatentSeek.**    We use the official code [4] provided by Li et al. (2025).

## A.4    LTPO IMPLEMENTATION DETAILS

**Prompt design.**    To ensure a fair comparison, LTPO uses the same prompt template as the Zero-Shot CoT-Unk for all experiments. See Appendix F for examples.

**Hyperparameter selection protocol.**    The performance of LTPO is dependent on several test-time hyperparameters, as detailed in Table 6. To determine these values, we employ a grid search methodology. For benchmarks with standard training splits, such as GSM8K and ASDiv-Aug, we hold out a validation set with 200 examples from the training data. For benchmarks without a training set (MATH-500, AIME2024, AIME2025), we use the GSM8K training set as a proxy validation environment to find a reasonable starting point, followed by minimal adjustments. While our framework is training-free, we acknowledge that finding the optimal set of hyperparameters for a new task requires this selection process. This represents a trade-off for the method's adaptability and robustness without requiring any model parameter updates.

To quantify this trade-off and evaluate the robustness of LTPO, we conduct an additional experiment using a single, fixed default hyperparameter configuration for all tasks, removing the need for per-task tuning. The results, shown in Table 7, reveal that even without task-specific tuning, LTPO remains highly effective.

---

[2] https://huggingface.co/xufangzhi/Genius_Magpie-25K_LLaMA3.1-8B-Instruct
[3] https://github.com/xuyige/SoftCoT
[4] https://github.com/bigai-nlco/LatentSeek

Table 6: Hyperparameters for LTPO. lr: learning rate.

| model | benchmark | # thought tokens | # steps | top-$k$ | sigma | sigma decay | lr |
|---|---|---|---|---|---|---|---|
| LLaMA-3.1-8B Instruct | GSM8K | 10 | 20 | 10 | 10 | 0.9 | 1e-2 |
| | MATH500 | 10 | 20 | 10 | 5 | 0.9 | 5e-4 |
| | ASDiv-Aug | 2 | 20 | 10 | 10 | 0.9 | 1e-2 |
| | AIME2024 | 8 | 10 | 10 | 4 | 0.9 | 4e-2 |
| | AIME2025 | 8 | 20 | 10 | 5 | 0.9 | 5e-3 |
| LLaMA-3.2-3B Instruct | GSM8K | 4 | 20 | 5 | 20 | 0.95 | 5e-3 |
| | MATH500 | 8 | 20 | 20 | 20 | 0.95 | 1e-4 |
| | ASDiv-Aug | 4 | 5 | 10 | 1 | 1.0 | 1e-3 |
| | AIME2024 | 4 | 20 | 10 | 10 | 0.95 | 1e-2 |
| | AIME2025 | 8 | 20 | 10 | 10 | 0.8 | 1e-3 |
| Qwen-2.5-7B Instruct | GSM8K | 8 | 20 | 10 | 20 | 0.95 | 1e-2 |
| | MATH500 | 8 | 20 | 10 | 20 | 0.95 | 5e-3 |
| | ASDiv-Aug | 5 | 20 | 5 | 10 | 0.9 | 1e-2 |
| | AIME2024 | 8 | 20 | 10 | 5 | 0.95 | 5e-2 |
| | AIME2025 | 10 | 10 | 10 | 20 | 0.95 | 1e-4 |
| Qwen-3-14B | GSM8K | 10 | 20 | 10 | 20 | 0.95 | 1e-3 |
| | MATH500 | 10 | 20 | 10 | 20 | 0.95 | 5e-3 |
| | ASDiv-Aug | 5 | 20 | 10 | 10 | 0.9 | 1e-2 |
| | AIME2024 | 4 | 20 | 10 | 8 | 0.9 | 4e-4 |
| | AIME2025 | 4 | 20 | 10 | 8 | 0.9 | 1e-2 |

Table 7: Robustness experiment with a fixed hyperparameter set on LLaMA-3.1-8B-Instruct. Reported in accuracy (%). The fixed set used is: # thought tokens = 8, # steps = 20, top-$k$ = 10, sigma = 5, sigma decay = 0.9, lr = 5e-3. The optimal results are in bold and the suboptimal ones are underlined.

| Method | GSM8K | MATH 500 | ASDiv Aug | AIME 2024 | AIME 2025 | Avg. |
|---|---|---|---|---|---|---|
| Zero-Shot CoT | 76.88 | 48.60 | 79.58 | 6.67 | 0.00 | 42.35 |
| SoftCoT | 80.36 | 39.80 | 87.57 | 0.00 | 0.00 | 41.55 |
| LatentSeek (Prompt 1) | 49.89 | **52.20** | 46.15 | 3.33 | 0.00 | 30.31 |
| LatentSeek (Prompt 2) | **83.70** | 47.40 | **90.66** | 3.33 | 0.00 | 45.02 |
| LTPO (Per-task Tuned) | 81.27 | 49.00 | 89.69 | **16.67** | **6.67** | **48.66** |
| LTPO (Fixed Default Config) | 80.29 | 45.80 | 88.05 | 13.33 | **6.67** | 46.83 |

The LTPO with a fixed hyperparamter set achieves an average accuracy of 46.83%, only a minor decrease from the 48.66% achieved with per-task tuning. Most importantly, it still decisively outperforms Zero-Shot CoT (42.35%), SoftCoT (41.55%), LatentSeek with prompt 1 (30.31%), and LatentSeek with prompt 2 (45.02%), and maintains its unique advantage on the difficult AIME benchmarks. This result strongly suggests that while per-task tuning can provide a final optimization boost, the core reasoning enhancement from LTPO is robust and can be achieved with a general-purpose configuration, affirming its practical utility.

# B    LIMITATION: THE DIVERGENCE OF CONFIDENCE AND CORRECTNESS

## B.1    QUALITATIVE ANALYSIS

To better understand the limitations of LTPO, we analyze its failure cases. A key failure mode arises from the nature of the confidence-based reward. In some instances, the optimization process successfully increases the model's confidence, but in a flawed reasoning direction, leading to a confidently incorrect answer.

This confirms that the intrinsic reward, while effective, is not a perfect proxy for correctness and highlights a known challenge for self-improving systems. We illustrate this divergence with a qualitative example from the MATH-500 dataset in Table 10. The model is tasked with finding the least positive integer multiple of 30 composed only of digits 0 and 2.

The LTPO loop optimizes for the highest reward, which corresponds to the model's internal confidence. In this case, the process converges on an incorrect solution (120) that achieved a higher reward (6.375) than the correct solution (2220), which had a lower reward (4.781).

The incorrect reasoning latches onto a flawed intermediate step ("Step 5: Multiplying 222 by 2 will make it divisible by 10...") and then attempts to modify it, leading to a series of logical errors. Despite being wrong, the step-by-step structure is fluent and assertive. The LTPO loop refines the latent thoughts to produce this confident, yet ultimately wrong, solution based on the initial error. In contrast, the correct reasoning is sound but produces a lower confidence score. This suggests the optimization can overfit to the model's internal sense of confidence, even when that confidence is misplaced.

Future work could explore incorporating uncertainty estimation or other signals to create more robust rewards that better align with logical correctness.

## B.2 THEORETICAL ANALYSIS

To analyze the relationship between confidence and correctness, it is productive to model the problem geometrically. The space of possible reasoning strategies can be conceptualized as a high-dimensional manifold. On this manifold, we can define two distinct scalar functions: a "confidence" function, which is perceptible and optimizable by the LLM at test time, and a "correctness" function, which represents the hidden ground truth. This geometric perspective provides a framework for understanding the dynamics of LTPO and the reasons for its successes and failures.

**Latent thought manifold.** LTPO optimizes a set of $K$ latent thought vectors, $\boldsymbol{H} = (\boldsymbol{h}_1, ..., \boldsymbol{h}_K)$, where each $\boldsymbol{h}_i \in \mathbb{R}^d$ and $d$ is the model's hidden dimension. The space of all possible sets of these vectors, $\boldsymbol{H} \in \mathbb{R}^{K \times d}$, can be viewed as a high-dimensional manifold, $\mathcal{H} = \mathbb{R}^{K \times d}$, where each point represents a unique reasoning strategy.

**Confidence landscape.** For any point $\boldsymbol{H}$, the model's policy $\pi(\cdot|\boldsymbol{H})$ samples actions (probes) to compute a reward. The objective of LTPO is to maximize the expected reward, which defines the confidence landscape: $J_{conf}(\boldsymbol{H}) = \mathbb{E}_{\boldsymbol{A} \sim \pi(\cdot|\boldsymbol{H})}[R(\boldsymbol{A})]$. This landscape is observable, and LTPO performs stochastic gradient ascent upon it.

**Correctness landscape.** We can define a theoretical function, $f_{corr}(\boldsymbol{A}) \to \{0, 1\}$, which is 1 if an action $\boldsymbol{A}$ leads to a correct final answer and 0 otherwise. While $f_{corr}$ itself is discrete, non-differentiable step function, we are interested in the expected correctness: $J_{corr}(\boldsymbol{H}) = \mathbb{E}_{\boldsymbol{A} \sim \pi(\cdot|\boldsymbol{H})}[f_{corr}(\boldsymbol{A})]$. This expectation is a smooth function of $\boldsymbol{H}$, and its gradient, $\nabla_{\boldsymbol{H}} J_{corr}(\boldsymbol{H})$, is well-defined via the policy gradient theorem. This landscape represents the "ground truth" we ultimately wish to optimize. However, it is fundamentally non-observable at test time, as determining the correctness of an answer requires an external verifier.

**Update dynamics.** LTPO navigates the latent thought manifold $\mathcal{H}$ by iteratively updating the parameters $\boldsymbol{H}$ to maximize the confidence objective. At each step $t$, the update follows the gradient of the confidence landscape, defined by the rule $\boldsymbol{H}^{(t+1)} = \boldsymbol{H}^{(t)} + \eta \nabla_{\boldsymbol{H}} J_{conf}(\boldsymbol{H}^{(t)})$, where $\eta$ is the learning rate. This dynamic can be modeled as a *gradient flow* in the continuous limit. Specifically, in the limit of a small learning rate $\eta \to 0$, the trajectory of the parameters $\boldsymbol{H}$ follows the *gradient flow* (Ordinary Differential Equation) defined by the expected reward:

$$\frac{d\boldsymbol{H}}{dt} = \nabla_{\boldsymbol{H}} J_{conf}(\boldsymbol{H}). \tag{12}$$

**Theorem B.1** *(The Gradient Alignment Condition). A gradient ascent step on the expected confidence landscape, $J_{conf}(\boldsymbol{H})$, also increases the expected correctness landscape, $J_{corr}(\boldsymbol{H})$, if and*

*only if the gradients of the two landscapes are positively aligned. Formally, this condition is:*

$$\nabla_{\boldsymbol{H}} J_{conf}(\boldsymbol{H}) \cdot \nabla_{\boldsymbol{H}} J_{corr}(\boldsymbol{H}) > 0. \tag{13}$$

*Proof:*

1. The LTPO update rule moves the latent thought vectors $\boldsymbol{H}$ in the direction of the estimated confidence gradient: $\Delta\boldsymbol{H} = \eta \cdot \nabla_{\boldsymbol{H}} J_{conf}(\boldsymbol{H})$, for a small learning rate $\eta > 0$.

2. The change in expected correctness from this step can be approximated by the directional derivative of $J_{corr}$ in the direction of the update. For an infinitesimal step (consistent with the *gradient flow* in Equation 12), this change is given by $\Delta J_{corr} \approx \nabla_{\boldsymbol{H}} J_{corr}(\boldsymbol{H}) \cdot \Delta\boldsymbol{H}$.

3. Substituting the expression for $\Delta\boldsymbol{H}$, we get $\Delta J_{corr} \approx \eta(\nabla_{\boldsymbol{H}} J_{corr}(\boldsymbol{H}) \cdot \nabla_{\boldsymbol{H}} J_{conf}(\boldsymbol{H}))$.

4. Since $\eta > 0$, the change in expected correctness, $\Delta J_{corr}$, is positive if and only if the dot product of the two gradients, $\nabla_{\boldsymbol{H}} J_{corr}(\boldsymbol{H}) \cdot \nabla_{\boldsymbol{H}} J_{conf}(\boldsymbol{H})$, is positive. This means the update step is productive (i.e., it increases correctness) only when the angle between the two gradients is acute ($< 90$ degrees).

**Theorem B.2** *(The "Confidently Incorrect" Trap). Let $\boldsymbol{H}_{trap} \in \mathcal{H}$ be a strict local maximizer of expected confidence landscape $J_{conf}$, such that $J_{corr}(\boldsymbol{H}_{trap}) < \epsilon$ for some small $\epsilon > 0$. Let $\mathcal{B}(\boldsymbol{H}_{trap}) \subset \mathcal{H}$ be the basin of attraction of $\boldsymbol{H}_{trap}$ under the gradient flow of $J_{conf}$ (Equation 12). For any initialization $\boldsymbol{H}^{(0)} \in \mathcal{B}(\boldsymbol{H}_{trap})$, the trajectory converges to $\boldsymbol{H}_{trap}$, resulting in a solution with expected correctness less than $\epsilon$.*

*Proof:*

1. To prove convergence to $\boldsymbol{H}_{trap}$, we construct a Lyapunov function. Consider the function $V(\boldsymbol{H}) = J_{conf}(\boldsymbol{H}_{trap}) - J_{conf}(\boldsymbol{H})$. Inside the neighborhood of $\boldsymbol{H}_{trap}$, since $\boldsymbol{H}_{trap}$ is a strict local maximizer, $V(\boldsymbol{H}) > 0$ for all $\boldsymbol{H} \neq \boldsymbol{H}_{trap}$ and $V(\boldsymbol{H}_{trap}) = 0$.

   Taking the time derivative of $J_{conf}(\boldsymbol{H})$ along the trajectory of the *gradient flow*:

   $$\frac{d}{dt} J_{conf}(\boldsymbol{H}) = \nabla_{\boldsymbol{H}} J_{conf}(\boldsymbol{H}) \cdot \frac{d\boldsymbol{H}}{dt}. \tag{14}$$

   Substituting the dynamics (Equation 12):

   $$\frac{d}{dt} J_{conf}(\boldsymbol{H}) = \|\nabla_{\boldsymbol{H}} J_{conf}(\boldsymbol{H})\|^2 \geq 0. \tag{15}$$

   This implies that $J_{conf}(\boldsymbol{H})$ is strictly increasing along any non-stationary trajectory.

   Consequently, $V(\boldsymbol{H})$ is strictly decreasing within the neighborhood of $\boldsymbol{H}_{trap}$. By Lyapunov's Stability Theorem (Lyapunov, 1992), the equilibrium point $\boldsymbol{H}_{trap}$ is *asymptotically stable*.

2. By the definition of the basin of attraction $\mathcal{B}(\boldsymbol{H}_{trap})$, for any initial condition $\boldsymbol{H}^{(0)} \in \mathcal{B}(\boldsymbol{H}_{trap})$, the limit of the flow is:

   $$\lim_{t \to \infty} \boldsymbol{H} = \boldsymbol{H}_{trap}. \tag{16}$$

   For the discrete-time algorithm used in LTPO (Equation 10), standard approximation theory for stochastic approximation (Robbins & Monro, 1951) guarantees that if step sizes $\eta_t$ satisfy $\sum_t \eta_t = \infty$ and $\sum_t \eta_t^2 < \infty$, the discrete iterates converge almost surely to the stable equilibrium points of the ODE (Equation 12 in our case). Assuming a sufficiently small constant $\eta$ and bounded variance, the system remains bounded within the basin and converges to a neighborhood of $\boldsymbol{H}_{trap}$.

3. We are interested in the correctness of the converged state. Since the trajectory converges to $\boldsymbol{H}_{trap}$, the correctness of the final state is:

   $$\lim_{t \to \infty} J_{corr}(\boldsymbol{H}) = J_{corr}(\lim_{t \to \infty} \boldsymbol{H}) = J_{corr}(\boldsymbol{H}_{trap}). \tag{17}$$

By the premise of the theorem, we have $J_{corr}(\boldsymbol{H}_{trap}) < \epsilon$. Therefore, the optimization process drives the latent thought vectors to a fixed point $\boldsymbol{H}_{trap}$ where the model is maximally confident ($\nabla_{\boldsymbol{H}} J_{conf}(\boldsymbol{H}_{trap}) = 0$) but the expected correctness is negligible ($J_{corr}(\boldsymbol{H}_{trap}) < \epsilon$). The system is mathematically trapped in a high-confidence, low-accuracy state.

## C  GENERALIZATION TO COMMONSENSE AND SYMBOLIC REASONING

In this section, we present the comprehensive experimental results supporting the generalization capabilities of LTPO discussed in Section 4.6. To assess whether the benefits of LTPO extend beyond mathematical reasoning, we evaluated our framework on two distinct tasks: StrategyQA (Geva et al., 2021) for commonsense reasoning and Date Understanding (Srivastava et al., 2023) from BIG-benchmark for symbolic reasoning.

The results, summarized in Table 8, demonstrate that LTPO consistently outperforms both the Zero-Shot CoT baseline and the training-based SoftCoT baseline across diverse domains.

**Commonsense reasoning.**  StrategyQA requires models to infer implicit reasoning steps to answer boolean questions (e.g., "Did aristotle use a laptop?"). LTPO demonstrates robust generalization to semantic reasoning tasks. On Qwen-2.5-7B-Instruct, LTPO achieves 73.09% accuracy, surpassing the Zero-Shot CoT baseline of 72.63%. Similarly, on LLaMA-3.1-8B-Instruct, LTPO improves upon the baseline (71.49%) to reach 72.18%. This indicates that the test-time optimization of latent thoughts maintains robust performance on natural language logic, providing consistent gains even on tasks distinct from mathematical reasoning.

**Symbolic reasoning.**  The Date Understanding task involves logical manipulation of time intervals (e.g., calculating relative dates). LTPO delivers its most significant non-mathematical improvements on this symbolic task. With Qwen-2.5-7B-Instruct, LTPO achieves 76.40% accuracy, outperforming Zero-Shot CoT (64.40%) by a substantial margin of +12.0%. It also surpasses SoftCoT (68.40%) by +8.0%. This trend holds for LLaMA-3.1-8B-Instruct, where LTPO (62.00%) outperforms both Zero-Shot CoT (55.60%) and SoftCoT (58.40%).

These experiments highlight a critical advantage of LTPO: adaptability. While training-based latent reasoning methods like SoftCoT can be brittle or underperform even with domain-specific training, LTPO leverage the model's intrinsic confidence landscape to discover optimal reasoning paths for each specific instance. This makes LTPO a highly robust framework capable of enhancing reasoning across mathematical, symbolic, and commonsense domains.

Table 8: Performance of LTPO vs. SoftCoT vs. Zero-Shot CoT on StrategyQA (commonsense) and DU (symbolic) across 2 models, reported in accuracy (%). "DU" indicates the Date Understanding dataset. The optimal results are in bold and the suboptimal ones are underlined.

| Model | Method | Commonsense StrategyQA | Symbolic DU | Avg. |
|---|---|---|---|---|
| LLaMA-3.1-8B-Instruct | Zero-Shot CoT | 71.49 | 55.60 | 63.55 |
| | SoftCoT | 67.34 | 58.40 | 62.87 |
| | **LTPO (Ours)** | **72.18** | **62.00** | **67.09** |
| Qwen-2.5-7B-Instruct | Zero-Shot CoT | 72.63 | 64.40 | 68.52 |
| | SoftCoT | 62.75 | 68.40 | 65.58 |
| | **LTPO (Ours)** | **73.09** | **76.40** | **74.75** |

## D    ANALYSIS OF OPTIMIZATION OVERHEAD VS. GENERATION SAVINGS

In Section 4.4, we reported that LTPO is faster than Zero-Shot CoT on challenging benchmarks, which may seem counter-intuitive given that LTPO introduces an iterative optimization loop at test time. Here, we provide a granular breakdown of the inference time to explain this phenomenon.

The total inference time consists of two parts: the optimization phase (unique to LTPO) and the final answer generation phase (common to all methods).

- **Optimization Overhead:** The optimization loop in LTPO is computationally efficient because it does not involve autoregressive decoding. For the AIME2024 benchmark with $T = 20$ steps, the optimization stage takes approximately **1.86 seconds** per problem.
- **Generation Savings:** The majority of the computational cost for reasoning tasks lies in the autoregressive generation of the final answer. We observe that optimizing latent thought tokens allows the model to reach the correct solution with a much more concise textual output.

Table 9 compares the average generated token length between Zero-Shot CoT and LTPO on the AIME2024 benchmark. Zero-Shot CoT tends to generate long, winding reasoning chains (averaging 2,886 tokens). In contrast, LTPO, guided by the optimized latent thoughts, generates significantly shorter paths to the solution (averaging 1,420 tokens).

Table 9: Comparison of average optimization time, generated token length, and total inference time per problem on the AIME2024 benchmark using LLaMA-3.1-8B-Instruct. The optimal results are in bold.

| Method | Opt. Time (s) | Avg. # Tokens | Total Time (s) |
|---|---|---|---|
| Zero-Shot CoT | **0.00** | 2886 | 62.59 |
| **LTPO (Ours)** | 1.86 | **1420** | **31.80** |

This reduction in generation length (near 50% reduction) far outweighs the minor cost of the optimization loop, resulting in a net speedup. This suggests that LTPO effectively compresses the necessary "thinking" into the latent vector space, reducing the need for verbose explicit reasoning.

## E    DETAILED ANALYSIS OF LATENT SPACE STABILITY

In Section 4.5.1 and Figure 2 (left), we observed that LTPO maintains robust performance as the number of thought tokens ($K$) increases from 1 to 16, whereas baseline methods like SoftCoT degrade significantly. This stability raises a fundamental question: does the addition of thought tokens provide additional learning signals, or does the method merely ignore them? Here, we analyze the mechanics behind this robustness, distinguishing between static projection and dynamic optimization.

### E.1    DYNAMIC OPTIMIZATION VS. STATIC PROJECTION

The divergence in performance stability between LTPO and SoftCoT stems from how latent thoughts are generated and utilized:

**Static Projection (SoftCoT).**    This approach relies on a static linear projection module, which is trained offline, to map the assistant model's thoughts into a fixed number of latent vectors. As the number of thought tokens increases, the static projection module struggles to produce high-quality representations without introducing noise. This results in the brittleness observed in Figure 2, where SoftCoT degrades sharply beyond $K = 4$.

**Dynamic Optimization (LTPO).**    In contrast, LTPO treats the $K$ latent thought tokens as dynamic parameters to be optimized. We do not rely on a fixed mapping. Instead, the online policy gradient updates (Equation 10) actively refine all $K$ vectors for each specific problem instance. This allows

the model to "prune" non-useful directions in the latent space and align the vectors to maximize the intrinsic confidence reward, regardless of the initialization or count.

### E.2 THE LATENT "WORKSPACE" HYPOTHESIS

We posit that LTPO treats the sequence of latent thought tokens as a dynamic "workspace". The stability of accuracy across varying $K$ demonstrates the method's ability to effectively manage this workspace capacity:

**Adaptability.** For a given problem, LTPO can find the best way to use the allotted tokens. If a simple problem only requires minimal latent reasoning (e.g., equivalent to 1 or 2 thought tokens), LTPO will effectively optimize 2 tokens and the remaining 14 tokens will be refined to a state that does not interfere with the prediction.

**Robustness to Redundancy.** Unlike static methods where redundant tokens introduce noise that disrupts the reasoning chain, LTPO's dynamic optimization ensures that additional capacity is either utilized for complex reasoning or neutralized.

Therefore, the stability of LTPO is not an indication that additional tokens lack signal. Rather, it validates that our test-time optimization effectively utilizes the allotted latent capacity, whether small or large, to find a high-confidence reasoning path without suffering from the degradation typical of static projection methods.

## F PROMPT TEMPLATES

In this section, we release the examples of AIME2024 for reference. Zero-Shot CoT-Unk and LTPO share the same prompt template, which is closely based on the one used by SoftCoT. We use reserved special tokens as the initial latent thought tokens.

In the following examples (Table 11 and Table 12), content under the "Answer" section is the reasoning generated by the LLM, and the content under the "Model Prediction" section is the final numerical answer that is extracted by our parser.

## G THE USE OF LARGE LANGUAGE MODELS

We acknowledge the use of the LLM, Gemini-2.5-Pro (Google, 2025), to help us polish the writing of this paper.

Table 10: Qualitative example of LTPO's confidence-correctness divergence. The model assigns a higher confidence-based reward to a fluent but incorrect reasoning path than to the correct solution, illustrating a key failure mode. The flawed intermediate step is marked with red color.

---

**QUESTION:**

What is the least positive integer multiple of 30 that can be written with only the digits 0 and 2?

**GROUND TRUTH:**

2220

**CONFIDENT BUT INCORRECT ANSWER:**

## Step 1: To find the least positive integer multiple of 30 that can be written with only the digits 0 and 2, we need to consider the properties of multiples of 30. A multiple of 30 must be divisible by both 2 and 3.
## Step 2: Since the number must be written with only the digits 0 and 2, it must be an even number, making it divisible by 2. Therefore, we only need to focus on making it divisible by 3.
## Step 3: To be divisible by 3, the sum of the digits of the number must be divisible by 3. The smallest possible number that meets this condition and is written with only the digits 0 and 2 is 222, as 2 + 2 + 2 = 6, which is divisible by 3.
## Step 4: However, 222 is not a multiple of 30. To make it a multiple of 30, we need to multiply it by the smallest factor that will make it divisible by 10 (to make it a multiple of 10) and then by 3 (to maintain divisibility by 3).
## Step 5: Multiplying 222 by 2 will make it divisible by 10, resulting in 444. However, 444 is not a multiple of 3. We need to find the next multiple of 3 that is also a multiple of 10.
## Step 6: The next multiple of 3 that is also a multiple of 10 is 666, as 6 + 6 + 6 = 18, which is divisible by 3. However, we can find a smaller multiple by considering the properties of multiples of 30.
## Step 7: A multiple of 30 can be expressed as 30n, where n is a positive integer. To find the smallest multiple of 30 that can be written with only the digits 0 and 2, we need to find the smallest n such that 30n is a multiple of 10 and has only the digits 0 and 2.
## Step 8: The smallest n that satisfies these conditions is n = 4, as 30 * 4 = 120, which is a multiple of 10 and has only the digits 0 and 2.
## Step 9: Therefore, the least positive integer multiple of 30 that can be written with only the digits 0 and 2 is 120.
\boxed{120}

**CORRECT ANSWER WITH LOWER CONFIDENCE:**

## Step 1: To find the least positive integer multiple of 30 that can be written with only the digits 0 and 2, we need to consider the properties of multiples of 30. A multiple of 30 must be divisible by both 2 and 3.
## Step 2: Since the number must be written with only the digits 0 and 2, it must be an even number, making it divisible by 2. Therefore, we only need to focus on making it divisible by 3.
## Step 3: To be divisible by 3, the sum of the digits of the number must be divisible by 3. The sum of the digits 0 and 2 is 2, which is not divisible by 3.
## Step 4: To make the sum of the digits divisible by 3, we need to add more digits that will make the total sum divisible by 3. The smallest number that can be formed using only the digits 0 and 2 and is divisible by 3 is 222.
## Step 5: However, 222 is not a multiple of 30. To make it a multiple of 30, we need to multiply it by 3, resulting in 666. But 666 is not a multiple of 30 either.
## Step 6: We need to find the smallest multiple of 222 that is also a multiple of 10 (since 30 = 2 * 3 * 5). The smallest multiple of 222 that is also a multiple of 10 is 2220.
## Step 7: Therefore, the least positive integer multiple of 30 that can be written with only the digits 0 and 2 is 2220.
\boxed{2220}

---

Table 11: Prompt Example 1 for LTPO on AIME2024.

**Input:**
Solve the following math problem efficiently and clearly:
- For simple problems (2 steps or fewer):
Provide a concise solution with minimal description.
- For complex problems (3 steps or more):
Use this step-by-step format:
## Step 1: [Brief calculations]
## Step 2: [Brief calculations]

...
IMPORTANT: Regardless of the approach, you MUST always put your final answer within $\boxed{{}}$.
PROBLEM: Jen enters a lottery by picking 4 distinct numbers from $S = \{1, 2, 3, \cdots, 9, 10\}$. 4 numbers are randomly chosen from $S$. She wins a prize if at least two of her numbers were 2 of the randomly chosen numbers, and wins the grand prize if all four of her numbers were the randomly chosen numbers. The probability of her winning the grand prize given that she won a prize is $\frac{m}{n}$ where $m$ and $n$ are relatively prime positive integers. Find $m + n$.
There are 8 special tokens that contain compressed latent reasoning information that might be useful for your reasoning.
If these tokens are useful for your case, you can use them as reference. If these tokens are not useful for your case, you can ignore them and focus back to solving the problem.
Here are the 8 special tokens: [UNK] [UNK] [UNK] [UNK] [UNK] [UNK] [UNK] [UNK]

**Answer:**
## Step 1: Calculate the total number of ways to choose 4 numbers from S. The total number of ways to choose 4 numbers from S is given by the combination formula: $\binom{10}{4} = \frac{10!}{4!(10-4)!} = 210$.
## Step 2: Calculate the number of ways to choose 4 numbers from S such that at least 2 of them match the randomly chosen numbers. We can break this down into two cases: exactly 2 matches and exactly 3 or 4 matches.
## Step 3: Calculate the number of ways to choose exactly 2 matches. The number of ways to choose 2 numbers from the 4 randomly chosen numbers is $\binom{4}{2} = 6$. The number of ways to choose 2 numbers from the remaining 6 numbers in S is $\binom{6}{2} = 15$. Therefore, the total number of ways to choose exactly 2 matches is $6 \cdot 15 = 90$.
## Step 4: Calculate the number of ways to choose exactly 3 or 4 matches. The number of ways to choose 3 numbers from the 4 randomly chosen numbers is $\binom{4}{3} = 4$. The number of ways to choose 1 number from the remaining 6 numbers in S is $\binom{6}{1} = 6$. Therefore, the total number of ways to choose exactly 3 matches is $4 \cdot 6 = 24$. The number of ways to choose 4 numbers from the 4 randomly chosen numbers is $\binom{4}{4} = 1$. Therefore, the total number of ways to choose exactly 4 matches is 1.
## Step 5: Calculate the total number of ways to choose 4 numbers from S such that at least 2 of them match the randomly chosen numbers. The total number of ways to choose 4 numbers from S such that at least 2 of them match the randomly chosen numbers is $90 + 24 + 1 = 115$.
## Step 6: Calculate the probability of winning the grand prize given that she won a prize. The probability of winning the grand prize given that she won a prize is $\frac{1}{115}$.
## Step 7: Simplify the fraction to find m and n. The fraction $\frac{1}{115}$ is already in its simplest form, so $m = 1$ and $n = 115$.
## Step 8: Calculate the sum of m and n. The sum of $m$ and $n$ is $1 + 115 = 116$.
The final answer is: $\boxed{116}$

**Model Prediction:**
116

**Evaluate:**
Correct!

Table 12: Prompt Example 2 for LTPO on AIME2024.

**Input:**
Solve the following math problem efficiently and clearly:
- For simple problems (2 steps or fewer):
Provide a concise solution with minimal description.
- For complex problems (3 steps or more):
Use this step-by-step format:
## Step 1: [Brief calculations]
## Step 2: [Brief calculations]
...
IMPORTANT: Regardless of the approach, you MUST always put your final answer within
$\boxed{{}}$.
PROBLEM: There exist real numbers $x$ and $y$, both greater than 1, such that $\log_x\left(y^x\right) = \log_y\left(x^{4y}\right) = 10$. Find $xy$.
There are 8 special tokens that contain compressed latent reasoning information that might be useful
for your reasoning.
If these tokens are useful for your case, you can use them as reference. If these tokens are not useful
for your case, you can ignore them and focus back to solving the problem.
Here are the 8 special tokens: [UNK] [UNK] [UNK] [UNK] [UNK] [UNK] [UNK] [UNK]

**Answer:**
## Step 1: We are given the equations $\log_x\left(y^x\right) = 10$ and $\log_y\left(x^{4y}\right) = 10$. We can rewrite
these equations using the properties of logarithms: $\log_x\left(y^x\right) = x\log_x(y) = 10$ and $\log_y\left(x^{4y}\right) = 4y\log_y(x) = 10$.
## Step 2: Let's denote $\log_x(y)$ as $a$ and $\log_y(x)$ as $b$. Then we have the equations $xa = 10$ and
$4yb = 10$.
## Step 3: From the first equation, we can express $a$ as $a = \frac{10}{x}$. Substituting this into the second
equation, we get $4y\log_y(x) = 4yb = 10$, which simplifies to $b = \frac{10}{4y}$.
## Step 4: We also know that $b = \frac{1}{a}$. Substituting $a = \frac{10}{x}$, we get $b = \frac{1}{\frac{10}{x}} = \frac{x}{10}$.
## Step 5: Equating the two expressions for $b$, we have $\frac{x}{10} = \frac{10}{4y}$. Solving for $x$ in terms of $y$, we get
$x = \frac{100}{4y} = \frac{25}{y}$.
## Step 6: Substituting $x = \frac{25}{y}$ into the equation $xa = 10$, we get $\frac{25}{y} \cdot a = 10$. Since $a = \frac{10}{x}$, we
have $\frac{25}{y} \cdot \frac{10}{x} = 10$.
## Step 7: Simplifying the equation, we get $\frac{250}{xy} = 10$. Solving for $xy$, we have $xy = \frac{250}{10} = 25$.
The final answer is: $\boxed{25}$

**Model Prediction:**
25

**Evaluate:**
Correct!

