# OpenReview forum: "Thinking on the Fly: Test-Time Reasoning Enhancement via Latent Thought Policy Optimization"
_ICLR.cc/2026/Conference — ICLR 2026 Poster_

### Official Review · Reviewer_JXR4 · 2025-10-31

**Soundness:** 2
**Presentation:** 2
**Contribution:** 4
**Rating:** 4
**Confidence:** 4

**Summary:**

The paper introduces LIPO, a test-time optimization framework that enhances LLM’s reasoning abilities without updating its parameters. It uses internal confidence as a reward to optimize latent thought, which is then integrated with the prompt to generate a solution. They empirically show robust performance on the reasoning benchmark.

**Strengths:**

This paper considers using internal reward to optimize the latent thought, which is a test-time method, and does not need additional data.

**Weaknesses:**

see questions.

**Questions:**

How to choose the variance hyperparameter during updating is not explained.
For confidence, there are many metrics to measure it, like entropy, KL over a uniform distribution. What is the advantage of using negative log probability?
The Monte Carlo estimation of the gradient has large variance, as it only uses one sample.
Although many papers use internal confidence as a reward, I'm not sure if maximizing confidence is really meaningful, as it will reduce the diversity of the model.
Why use greedy decoding to generate the final answer? The technical report of Qwen mentions the optimal parameters, like temperature = 0.7, top-p = 0.8, and top-k = 20. Also, for the generation length, can you make it larger, as the Qwen3-14B can achieve good performance like 79.3 on AIME24 and 70.4 on AIME25, when the generation length equals 64k. Using a small generation length on these challenge reasoning tasks (AIME) is meaningless, as most methods didn’t complete the generation, achieving very low scores.
Is the optimal number of thought tokens and number of iterations the same for different reasoning tasks and different models?
Can you compare more baselines? Such as Seek in the Dark: Reasoning via Test-Time Instance-Level Policy Gradient in Latent Space or SLOT: Sample-specific Language Model Optimization at Test-time, which is also test-time for latent optimization.
Why is LTPO faster than zero-thot COT, as it needs optimization, but later does not need optimization?

---

> ### Author Response · Authors · 2025-11-20
> **Thanks for the review - Response to Reviewer JXR4 (Part I)**
>
> Dear Reviewer JXR4,
>
> We greatly appreciate your thorough and constructive comments, many of which have helped improve our paper. Both the paper and appendix have been updated extensively to include as much detail as possible. Please see our point-to-point response to your concerns below.
>
> > Q1: How to choose the variance hyperparameter during updating is not explained.
>
> Thank you for pointing out the need for greater clarity regarding the selection of the variance hyperparameter. In our implementation, the variance determines the exploration radius within the continuous latent space during the optimization process. We manage this hyperparameter through two mechanisms:
>
> **1. Selection Protocol (Grid Search):** As detailed in **Appendix A.4**, we determine the initial sigma via grid search.
>
> - For datasets with training splits (e.g., GSM8K), we utilize a held-out validation set.
> - For benchmarks without training sets (e.g., AIME, MATH-500), we use the GSM8K training set as a proxy validation environment to establish a baseline, followed by minimal adjustments.
> - The specific values used for each model and benchmark are listed in **Table 6**.
>
> **2. Decay Schedule:** To balance exploration and exploitation, we apply a decay factor (typically $0.9$ or $0.95$, see **Table 6**) at each step. This allows for broad exploration of the latent thought manifold in early iterations and finer refinement in later stages.
>
> **Robustness of the Variance Parameter:** Crucially, our experiments indicate that LTPO is not overly sensitive to the precise initialization of $\sigma$. As demonstrated in **Table 7**, we conducted a "Robustness experiment" using a **single fixed configuration** (initial $\sigma=5$, decay $=0.9$) across all benchmarks. This fixed setting achieved an average accuracy of 46.83%, which is comparable to the per-task tuned performance of 48.66% and still significantly outperforms baselines like SoftCoT (41.55%) and LatentSeek (45.02%).
>
> > Q2: For confidence, there are many metrics to measure it, like entropy, KL over a uniform distribution. What is the advantage of using negative log probability? The Monte Carlo estimation of the gradient has large variance, as it only uses one sample. Although many papers use internal confidence as a reward, I'm not sure if maximizing confidence is really meaningful, as it will reduce the diversity of the model.
>
> Thank you for your thoughtful questions and feedback. These points are at the core of our method, and we appreciate the opportunity to clarify our design choices and their empirical validation.
>
> **1. Choice of Confidence Metric: Top-k Log-Probability vs. Entropy**
>
> You are correct that other metrics exist. We chose the mean negative log-probability of the top-$k$ tokens for following key reasons:
>
> - **Alignment with Self-Certainty:** Recent works such as [1] and [2] have demonstrated that "Self-Certainty" (often measured via entropy or KL divergence) strongly correlates with reasoning correctness. Our metric serves as a computationally efficient proxy for this concept. Since a confident model places the vast majority of its probability mass on the top few tokens, maximizing the log-probability of the Top-$k$ tokens effectively minimizes the entropy of the distribution (i.e., maximizes Self-Certainty) without the computational overhead of processing the full vocabulary tail.
> - **Gradient Quality for Optimization:** Unlike post-hoc selection (where entropy is often used), LTPO performs online optimization via policy gradients. The gradient of the log-probability ($\nabla \log p$) is sharper and more direct for reinforcing specific latent thought vectors than the gradient of full entropy, which couples all vocabulary logits. By focusing on the Top-$k$ "head" of the distribution, we provide the optimizer with a clear, noise-free directional signal to sharpen the model's intent.
> - **Robustness to Tail Noise:** As noted in our sensitivity analysis (Table 4), performance is optimal with a small-to-medium $k$ ($k \le 10$). This confirms that the critical signal for reasoning confidence lies in the highest-probability tokens. Including the long tail (as full entropy does) introduces variance from thousands of irrelevant tokens, which can destabilize the test-time RL loop.
>
> [1] Kang, Zhewei, et al. "Scalable Best-of-N Selection for Large Language Models via Self-Certainty" AI for Math Workshop at ICML (2025).
>
> [2] Fu, Yichao, et al. "Deep Think with Confidence" arXiv preprint arXiv:2508.15260 (2025).

---

> ### Author Response · Authors · 2025-11-20
> **Thanks for the review - Response to Reviewer JXR4 (Part II)**
>
> **2. Variance of the Monte Carlo Gradient Estimation**
>
> This is an excellent point. Single-sample Monte Carlo estimation typically suffers from high variance. However, in the specific context of LTPO, this variance is mitigated by three factors:
>
> - **Local Optimization via Perturbation:** Our policy is a multivariate Gaussian centered at the current state $H^{(t)}$ with a decaying variance $\sigma^2$ (Equation 2). This effectively acts zero-order optimization in a local region. Since the perturbation $\epsilon$ is small and local, the reward difference is less volatile than in global search.
> - **Dense Reward Signal:** Our reward is intrinsic and computed directly from the output distribution (logits) at every optimization step, rather than a sparse "correct/incorrect" signal at the end of generation. This dense signal helps stabilize the gradient direction even with a single sample.
> - **Empirical Stability:** Despite the theoretical concern, our experiments demonstrate high stability. As shown in **Section 4.5.1** and **Appendix E**, the accuracy remains robust across different numbers of thought tokens, and Table 7 shows that a fixed set of hyperparameters generalizes well across different datasets, indicating that the gradient estimates are sufficiently accurate to guide the optimization.
>
> **3. Confidence Maximization and Model Diversity**
>
> You've raised an important conceptual point about the potential downsides of maximizing confidence, such as reducing diversity. However, we argue that for **logical and mathematical reasoning** (the focus of this paper), maximizing confidence is both desirable and necessary, for the following reasons:
>
> - **Reasoning vs. Creativity:** While diversity is crucial for creative generation (e.g., writing poetry), reasoning tasks like GSM8K and AIME typically have a single correct derivation path or at least a single correct final answer. In this domain, "diversity" in the output distribution often manifests as uncertainty or hallucination. Maximizing confidence effectively acts as "sharpening" the model's focus onto its most probable reasoning path, reducing the likelihood of straying into incorrect logic.
> - **Instance-Specific Optimization:** Crucially, LTPO is a test-time method that optimizes latent thoughts for a single specific problem instance. We are not updating the model's permanent weights. Therefore, we are not lobotomizing the model's general diversity or capabilities for future tasks. We are merely collapsing the uncertainty for the current math problem to ensure the most rigorous logical chain is followed.
> - **Exploration is Internal:** While we minimize diversity in the final output (by maximizing confidence), we actually increase exploration during the optimization process via the stochastic policy perturbation. This allows the model to explore the latent space to find a basin of high confidence (and hopefully high correctness) before committing to a generation.
> - **Validation:** Our results on the AIME benchmarks (Table 1) strongly support this hypothesis. On these hard tasks, base models are "diverse" but wrong (collapsing to 0-10% accuracy). By forcing the model to find a state of high confidence via LTPO, we recover significant performance (improving to 16.67%), suggesting that high-confidence states in the latent space correlate better with correct reasoning than the default high-entropy states.

---

> ### Author Response · Authors · 2025-11-20
> **Thanks for the review - Response to Reviewer JXR4 (Part III)**
>
> > Q3: Why use greedy decoding to generate the final answer? The technical report of Qwen mentions the optimal parameters, like temperature = 0.7, top-p = 0.8, and top-k = 20.
>
> While we acknowledge that the Qwen technical report recommends sampling parameters (e.g., temperature=0.7) for creative generation or open-ended chat, reasoning benchmarks require a different approach.
>
> We utilized greedy decoding for three primary reasons:
>
> - Greedy decoding is the standard approach for evaluating the Pass@1 metric in reasoning tasks. This ensures our results are reproducible and directly comparable to foundational literature. For instance, seminal works such as Chain-of-Thought Prompting [1] and Zero-Shot Reasoners [2] adopted the greedy decoding approach for their experiments. In addition, both SoftCoT [3] and LatentSeek [4] evaluate using a single reasoning path with greedy decoding, and report Pass@1 accuracy.
> - While tasks requiring creativity benefit from the variance introduced by higher temperatures, reasoning tasks aim to assess whether the model's most probable, first-attempt solution is correct.
> - Recent research supports this distinction. Song et al. [5] observed a notable performance gap in their analysis, finding that greedy decoding generally outperforms the average score of sampling methods on most evaluated benchmarks, particularly in mathematics and code generation.
>
> [1] Wei, Jason, et al. "Chain-of-Thought Prompting Elicits Reasoning in Large Language Models" NeurIPS (2022).
>
> [2] Kojima, Takeshi, et al. "Large Language Models are Zero-Shot Reasoners" NeurIPS (2022).
>
> [3] Xu, Yige, et al. "SoftCoT: Soft Chain-of-Thought for Efficient Reasoning with LLMs" ACL (2025).
>
> [4] Li, Hengli, et al. "Reasoning via Test-Time Instance-Level Policy Gradient in Latent Space" arXiv preprint arXiv:2505.13308 (2025).
>
> [5] Song, Yifan, et al. "The Good, The Bad, and The Greedy: Evaluation of LLMs Should Not Ignore Non-Determinism" ACL (2025).
>
> > Q4: Also, for the generation length, can you make it larger, as the Qwen3-14B can achieve good performance like 79.3 on AIME24 and 70.4 on AIME25, when the generation length equals 64k. Using a small generation length on these challenge reasoning tasks (AIME) is meaningless, as most methods didn’t complete the generation, achieving very low scores.
>
> We agree that restricting the generation length to 4096 tokens significantly bottlenecks the reasoning capabilities of advanced models like Qwen3-14B, particularly on challenge-level benchmarks like AIME where extensive "Chain of Thought" expansion is required.
>
> Per your suggestion, we conducted a new set of experiments using Qwen3-14B with the maximum output tokens extended to **64,000**. This setup allows the model to fully utilize its test-time computing potential.
>
> **Results with Extended Generation Length (64k)**
>
> As the reviewer mentioned, the performance of the baseline (Zero-Shot CoT) improved dramatically compared to the reported results in the paper (from 10% to 75% average). However, crucially, **LTPO continues to outperform the strong Zero-Shot CoT baseline** even in this high-performance, long-context regime.
>
> Below is a summary of the new results, which we have added to the paper in **Section 4.7**:
>
> Table: Performance of LTPO vs. Zero-Shot CoT on AIME benchmarks with Qwen3-14B, reported in accuracy (%). The max output tokens is set to 64000. The optimal results are in bold.
>
> | **Method**      | **AIME24** | **AIME25** | **Avg.**  |
> |:----------------|-------------:|-------------:|----------:|
> | Zero-Shot CoT   | 80.00        | 70.00        | 75.00     |
> | **LTPO (Ours)** | **83.33**    | **76.67**    | **80.00** |
>
> **Key Findings**
>
> - **State-of-the-Art Baseline:** With the generation length extended to 64k, the Zero-Shot CoT baseline achieves an impressive 75.00% average accuracy, confirming that this setup represents the "strong reasoning" regime the reviewer mentioned (a massive increase from the 10% accuracy observed under standard settings in Table 1 of our paper).
> - **LTPO Adds Value to Strong Reasoners:** Even atop this extremely high baseline, LTPO achieves a further performance boost, reaching an average accuracy of **80.00%** (+5.00%). Notably, LTPO pushes the AIME2024 score to 83.33% and AIME2025 to 76.67%.
> - **Consistency:** This improvement demonstrates that LTPO is not merely a compensation mechanism for weaker models (like LLaMA-3.1-8B). Instead, the optimization of latent thoughts acts as a complementary enhancement to long-context explicit reasoning: even when the model can “think” extensively in natural language, optimizing the initial latent thought vectors provides a better starting point for the reasoning trajectory.
>
> We have included these updated results in our paper. We appreciate the reviewer for guiding us toward this stronger empirical validation.

---

> ### Author Response · Authors · 2025-11-20
> **Thanks for the review - Response to Reviewer JXR4 (Part IV)**
>
> > Q5: Can you compare more baselines? Such as Seek in the Dark: Reasoning via Test-Time Instance-Level Policy Gradient in Latent Space or SLOT: Sample-specific Language Model Optimization at Test-time, which is also test-time for latent optimization.
>
> We thank the reviewer for suggesting these relevant baselines. We agree that comparing against test-time latent optimization methods is critical to contextualize our contribution.
>
> Comparison with LatentSeek (Li et al., 2025) Per your suggestion, we have added it as a core baseline. We evaluated it using the official codebase on all benchmarks.
>
> As shown in our updated **Table 1 (Section 4.2)** and **Table 3 (Section 4.4)**, the comparison reveals a distinct trade-off profile (to facilitate a direct comparison, we also copy and present the data in the below tables):
>
> - **Robustness on Complex Tasks:** This is the primary differentiator. While LatentSeek is competitive on standard tasks (e.g., slightly outperforming LTPO on GSM8K with Qwen-2.5-7B: 89.54% vs. 88.17%), it struggles significantly on challenging competition math.
>     - On **AIME 2024** (LLaMA-3.1-8B), LatentSeek scores **3.33%** (best prompt), whereas LTPO achieves **16.67%**.
>     - On **AIME 2025** (Qwen-2.5-7B), LatentSeek collapses to **0.00%** (Prompt 1), while LTPO maintains robustness at **13.33%**.
>     - Overall, LTPO achieves the **highest average accuracy** across all 5 benchmarks for both tested models (e.g., **56.79%** vs. LatentSeek's best of **52.91%** on Qwen-2.5-7B).
> - **Computational Efficiency:** A critical limitation of LatentSeek is its reliance on full autoregressive decoding at each optimization step.
>     - As shown in Table 3, LatentSeek requires **872-1002 seconds** per problem on AIME 2024.
>     - In contrast, LTPO utilizes a dense reward signal without intermediate text generation, taking only **31.8 seconds** per problem, an improvement of over **25x in speed**.
>
> Regarding SLOT, we focused our comparison on LatentSeek as it represents the most direct state-of-the-art baseline for _instance-level policy gradient in latent space_, which is the closest methodological counterpart to LTPO. Given that LatentSeek (a more recent method) struggles with the computational overhead of test-time optimization on reasoning tasks, the efficiency advantages of LTPO over this class of methods are clearly established by the comparison provided in **Section 4.4**.
>
> **Table 1**: Performance of LTPO vs. LatentSeek (Prompt 1 & Prompt 2) vs. Zero-Shot CoT across two models and five reasoning benchmarks, reported in accuracy (%). The optimal results are in **bold** and the suboptimal ones are in _italic_. The * symbol indicates the accuracy is reported by the original LatentSeek paper.
>
> | **Model** | **Method** | **GSM8K** | **MATH500** | **ASDiv-Aug** | **AIME2024** | **AIME2025** |
> |--|--|--:|--:|--:|--:|--:|
> | LLaMA-3.1-8B-| Zero-Shot CoT          | 76.88     | 48.60       | 79.58     | 6.67      | 0.00     |
> | Instruct     | LatentSeek* (Prompt 1) | 78.54     | **54.60**   | -         | _10.00_   | -        |
> |              | LatentSeek* (Prompt 2) | **83.70** | 47.00       | -         | 6.67      | -        |
> |              | LatentSeek (Prompt 1)  | 53.15     | _52.20_     | 46.15     | 3.33      | 0.00     |
> |              | LatentSeek (Prompt 2)  | **83.70** | 47.40       | **90.66** | 3.33      | 0.00     |
> |              | **LTPO (Ours)**        | _81.27_   | 49.00       | _89.69_   | **16.67** | **6.67** |
> | Qwen-2.5-7B- | Zero-Shot CoT          | 88.55     | 72.40       | 91.71     | 10.00     | _10.00_  |
> | Instruct     | LatentSeek* (Prompt 1) | _89.46_   | **75.60**   | -         | _13.33_   | -        |
> |              | LatentSeek* (Prompt 2) | 85.06     | 57.60       | -         | _13.33_   | -        |
> |              | LatentSeek (Prompt 1)  | **89.54** | _73.60_     | 91.43     | 10.00     | 0.00     |
> |              | LatentSeek (Prompt 2)  | 86.20     | 57.80       | _91.81_   | 6.67      | 6.67     |
> |              | **LTPO (Ours)**        | 88.17     | _73.60_     | **92.20** | **16.67** | **13.33**|
>
> **Table 2**: Average inference time per problem on GSM8K, AIME2024, and AIME2025 with LLaMA-3.1-8B-Instruct, reported in seconds. We set the RL steps $T = 20$ for LTPO, and $T = 10$ for LatentSeek to follow the setting in the original paper. The optimal results are in **bold** and the suboptimal ones are in _italic_.
>
> | **Method** | **GSM8K** | **AIME2024** | **AIME2025** | **Avg.**  |
> |--|--:|--:|--:|--:|
> | Zero-Shot CoT         | _5.81_    | 62.59        | 55.60        | 41.33     |
> | SoftCoT               | 5.91      | **26.01**    | _35.03_      | _22.32_   |
> | LatentSeek (Prompt 1) | 48.99     | 872.47       | 708.40       | 543.29    |
> | LatentSeek (Prompt 2) | 61.83     | 1002.30      | 512.43       | 525.52    |
> | **LTPO (Ours)**       | **5.69**  | _31.80_      | **24.89**    | **20.79** |

---

> ### Author Response · Authors · 2025-11-27
> **Thanks for the review - Response to Reviewer JXR4 (Part V)**
>
> > Q6: Is the optimal number of thought tokens and number of iterations the same for different reasoning tasks and different models?
>
> Thank you for the question. To answer directly: No, the optimal numbers are not identical. Our validation (detailed in Table 6) shows they vary by task and model. However, our key finding is that LTPO is highly robust and not sensitive to these hyperparameters. This is a crucial advantage of our method. As shown in Table 7, we ran LTPO with a single, fixed default configuration across all benchmarks. Even without any per-task tuning, this fixed-config LTPO (46.83% avg) decisively outperforms all baselines, including Zero-Shot CoT (42.35%), SoftCoT (41.55%), and LatentSeek (45.02%). It also maintains its significant advantage on the difficult AIME benchmarks.
>
> > Q7: Why is LTPO faster than Zero-Shot CoT, as it needs optimization, but later does not need optimization?
>
> Thank you for this important question. It points to a crucial, and perhaps counter-intuitive, aspect of our method's efficiency. You're correct that LTPO introduces an optimization loop, which intuitively seems like an added computational cost. However, the key insight is that this optimization loop is computationally cheap, as it **doesn't involve any explicit token generation** (Section 3.2).
>
> To break down the execution time: on the highly challenging AIME2024 benchmark, the average inference time per problem for Zero-Shot CoT and LTPO is **62.59s** and **31.80s**, respectively. For LTPO, the optimization stage ($T=20$ RL steps) only takes around **1.86s** per problem. Thus, the main computational overhead for both methods comes from the final answer generation.
>
> A closer analysis shows that Zero-Shot CoT tends to generate a much longer answer than LTPO. Specifically, the average number of tokens generated per problem for Zero-Shot CoT and LTPO is **2886** and **1420**, respectively. This significant reduction in output length is why LTPO is faster overall, despite including the latent thought token optimization phase. To facilitate a direct comparison, we present the data in the following table:
>
> **Table:** Comparison of average optimization time, generated token length, and total inference time per problem on the AIME2024 benchmark using LLaMA-3.1-8B-Instruct. The optimal results are in bold.
> |**Method**|**Opt. Time (s)**|**Avg. # Tokens**|**Total Time (s)**|
> |:--|--:|--:|--:|
> |Zero-Shot CoT|**0.00**|2886|62.59|
> |**LTPO (Ours)**|1.86|**1420**|**31.80**|
>
> We have updated **Section 4.4** and **Appendix D** to include this detailed analysis.

---

### Official Review · Reviewer_Xm7C · 2025-10-31

**Soundness:** 3
**Presentation:** 3
**Contribution:** 2
**Rating:** 4
**Confidence:** 2

**Summary:**

The paper introduces Latent Thought Policy Optimization (LTPO), a framework for improving the reasoning ability of Large Language Models at test-time. The core idea is concatenating the prompt with K tokens as latent thoughts and improving the latent representation via policy gradient with confidence as reward, derived from the model’s output probabilities on these latent tokens.
The initial latent tokens are set to be the embedding of [THINK] token and policy is conditional Gaussian with controlled variance.
The method is evaluated on five reasoning benchmarks: GSM8K, MATH-500, ASDiv-Aug, AIME2024, and AIME2025. The results outperform Zero-Shot CoT, CoT-Unk, and SoftCoT baselines on above benchmarks.

**Strengths:**

- Clear Motivation and Problem Framing: The paper is well-motivated, presenting a clear method for test-time latent space optimization.
- The proposed algorithm LTPO is sound, lightweight with broad applicability.
- The empirical result shows robust improvement over several latent reasoning baselines.

**Weaknesses:**

1. Missing Comparison to Highly Relevant Contemporaneous Work. The major weakness is missing comparison to highly relevant work LatentSeek (Li et al., 2025). Both frameworks introduce a near-identical method: test-time optimization of latent representations using a policy gradient to enhance reasoning. The primary distinction lies in the reward function. LTPO proposes a logit-based confidence score specifically to avoid text generation during the optimization loop. In contrast, LatentSeek requires decoding the entire sequence at each step. Given that LTPO's main claimed advantage is its efficiency compared to LatentSeek, a direct empirical comparison and trade-off discussion against LatentSeek is essential to validate the paper's central contribution.
2. Questionable Justification for Using a Policy Gradient. The paper chooses R that is "non-differentiable with respect to the input latent vectors", which in turn necessitated the use of a policy optimization method like REINFORCE. This is technically true as top-k is involved in the loss calculation, and the paper conduct ablation study that shows small k is necessary for good performance. However, it fails to justify this fundamental design choice over a simpler, fully-differentiable alternative. For instance, the goal of "improving confidence" could be achieved by directly minimizing the entropy of the output distribution using standard, and typically more stable, backpropagation. And with backpropagation, the dependence on k might be different and might offer a more stable and efficient path to optimizing the latent tokens. The paper should have included a discussion on the design choice as policy gradient method provides higher variance and design bias/complexity.

**Questions:**

1. Could you provide an empirical comparison with LatentSeek (Li et al., 2025)? Given the similar test-time optimization paradigm, a detailed discussion is necessary to distinguish the pros and cons of both method.
2. Could you justify the choice of the non-differentiable top-k reward function? This choice necessitates using a policy gradient, and it would be helpful to understand why this was preferred over a fully-differentiable metric.

I will raise my score once any of the questions above are properly addressed.

---

> ### Author Response · Authors · 2025-11-20
> **Thanks for the review - Response to Reviewer Xm7C (Part I)**
>
> Dear Reviewer Xm7C,
>
> We greatly appreciate you read our paper very carefully and your informative feedback, which has helped improve our paper. Both the paper and appendix have been updated extensively to include as much detail as possible. Please see our response to your concerns.
>
> > Q1: Could you provide an empirical comparison with LatentSeek (Li et al., 2025)? Given the similar test-time optimization paradigm, a detailed discussion is necessary to distinguish the pros and cons of both method.
>
> Thank you for this essential suggestion.
>
> We agree that a direct comparison with LatentSeek [1] is critical. We have now run this comparison, and the new results strongly validate our paper's central contribution: **LTPO's confidence-based reward provides a superior trade-off, achieving both better robustness and a dramatic 25x efficiency gain**.
>
> Here is a summary of the new empirical results, which we have added to the paper in **Section 4.2** and **Section 4.4**:
>
> ### 1. Accuracy and Robustness Comparison
>
> Our new experiments show that while LatentSeek is competitive on standard benchmarks, it exhibits the same brittleness as other baselines on complex tasks. In contrast, LTPO demonstrates significant robustness and superior performance on the challenging AIME benchmarks.
>
> **Table 1**: Performance of LTPO vs. LatentSeek (Prompt 1 & Prompt 2) vs. Zero-Shot CoT across two models and five reasoning benchmarks, reported in accuracy (%). The optimal results are in **bold** and the suboptimal ones are in _italic_. The '*' symbol indicates the accuracy is reported by the original LatentSeek paper.
>
> | **Model**    | **Method**             | **GSM8K** | **MATH500** | **ASDiv-Aug** | **AIME2024** | **AIME2025** |
> |--------------|------------------------|----------:|------------:|--------------:|-------------:|-------------:|
> | LLaMA-3.1-8B-| Zero-Shot CoT          | 76.88     | 48.60       | 79.58         | 6.67         | 0.00         |
> | Instruct     | LatentSeek* (Prompt 1) | 78.54     | **54.60**   | -             | _10.00_      | -            |
> |              | LatentSeek* (Prompt 2) | **83.70** | 47.00       | -             | 6.67         | -            |
> |              | LatentSeek (Prompt 1)  | 53.15     | _52.20_     | 46.15         | 3.33         | 0.00         |
> |              | LatentSeek (Prompt 2)  | **83.70** | 47.40       | **90.66**     | 3.33         | 0.00         |
> |              | **LTPO (Ours)**        | _81.27_   | 49.00       | _89.69_       | **16.67**    | **6.67**     |
> | Qwen-2.5-7B  | Zero-Shot CoT          | 88.55     | 72.40       | 91.71         | 10.00        | _10.00_      |
> | Instruct     | LatentSeek* (Prompt 1) | _89.46_   | **75.60**   | -             | _13.33_      | -            |
> |              | LatentSeek* (Prompt 2) | 85.06     | 57.60       | -             | _13.33_      | -            |
> |              | LatentSeek (Prompt 1)  | **89.54** | _73.60_     | 91.43         | 10.00        | 0.00         |
> |              | LatentSeek (Prompt 2)  | 86.20     | 57.80       | _91.81_       | 6.67         | 6.67         |
> |              | **LTPO (Ours)**        | 88.17     | _73.60_     | **92.20**     | **16.67**    | **13.33**    |

---

> ### Author Response · Authors · 2025-11-20
> **Thanks for the review - Response to Reviewer Xm7C (Part II)**
>
> ### 2. Efficiency Comparison
>
> The reviewer correctly identified that one of the distinction is the reward function. Our analysis shows this design choice has a critical impact on performance. LatentSeek's reliance on full text decoding at each optimization step makes it computationally prohibitive.
>
> **Table 2**: Average inference time per problem on GSM8K, AIME2024, and AIME2025 with LLaMA-3.1-8B-Instruct, reported in seconds. We set the RL steps $T = 20$ for LTPO, and $T = 10$ for LatentSeek to follow the setting in the original paper. The optimal results are in **bold** and the suboptimal ones are in _italic_.
>
> | **Method**            | **GSM8K** | **AIME2024** | **AIME2025** | **Avg.**  |
> |-----------------------|----------:|-------------:|-------------:|----------:|
> | Zero-Shot CoT         | _5.81_    | 62.59        | 55.60        | 41.33     |
> | SoftCoT               | 5.91      | **26.01**    | _35.03_      | _22.32_   |
> | LatentSeek (Prompt 1) | 48.99     | 872.47       | 708.40       | 543.29    |
> | LatentSeek (Prompt 2) | 61.83     | 1002.30      | 512.43       | 525.52    |
> | **LTPO (Ours)**       | **5.69**  | _31.80_      | **24.89**    | **20.79** |
>
> LTPO is over **25 times faster** than LatentSeek on average, even while using _twice_ the number of optimization steps ($T=20$ for LTPO vs. $T=10$ for LatentSeek). This confirms that LTPO is a practical and "on-the-fly" framework, whereas LatentSeek is not.
>
> ### Conclusion
>
> This comparison clarifies the pros and cons:
>
> **LatentSeek:**
> - **Pro:** Validates the test-time policy gradient paradigm.
> - **Con:** Impractically slow (over 25x slower) due to its full text decoding at each RL step.
> - **Con:** Brittle; performance collapses on challenging AIME tasks.
>
> **LTPO (Ours):**
> - **Pro:** Highly efficient. Our novel, confidence-based reward makes it practical for real-world use.
> - **Pro:** Robust. It is shown to provide substantial gains on highly challenging benchmarks, demonstrating superior reasoning.
> - **Con:** As noted in our paper (Appendix B), the confidence reward is a proxy and can, in some cases, diverge from correctness.
>
> In summary, this new data validates our core contribution: LTPO's confidence-based reward function is the key innovation that makes test-time latent optimization both **practical and robust**.
>
> [1] Li, Hengli, et al. "Seek in the Dark: Reasoning via Test-Time Instance-Level Policy Gradient in Latent Space" arXiv preprint arXiv:2505.13308 (2025).

---

> > ### Author Response · Authors · 2025-11-26
> > **Thanks for the review - Response to Reviewer Xm7C (Part III)**
> >
> > > Q2: Could you justify the choice of the non-differentiable top-k reward function? This choice necessitates using a policy gradient, and it would be helpful to understand why this was preferred over a fully-differentiable metric.
> >
> > Thank you for this insightful question, which targets a key design choice of our LTPO framework.
> >
> > The reviewer suggests backpropagation would be "more stable". However, in our specific setting, this is unlikely to be the case.
> >
> > - **Our method optimizes input embeddings, not model weights.** LTPO optimizes the latent thought vectors $H$, which are inputs to the LLM.
> > - **Backpropagation through a deep LLM to update inputs is notoriously unstable.** Optimizing input-space vectors via gradients that must flow backward through dozens of transformer layers is a very different task from training model weights. This process is often plagued by vanishing/exploding gradients and highly complex loss landscapes, making stable optimization difficult, especially at the scene of test-time scaling.
> > - **Our Policy Gradient (PG) approach avoids this instability.** By using a PG method like REINFORCE, we treat the frozen LLM as a black-box reward function. The gradient update $\eta \cdot R(\cdot) \frac{\epsilon^{(t)}}{\sigma^{2}}$ (Equation 10) does not require differentiating through the LLM. This "zero-order" optimization-style is more stable for tuning the input latent thought vectors $H$.
> >
> > This approach is also validated by recent work. As the reviewer notes, recent works like [1] and [2], also build on this core intuition. While they primarily use confidence for _post-hoc_ selection , LTPO is distinct in using it as a dense, online reward signal. The fact that this principle (using top-k confidence as a proxy for quality) is now being successfully applied across different paradigms (post-hoc selection and our online optimization) suggests it is a robust and valid approach. Furthermore, our finding in Section 4.5.2 (to track the best reward found during exploration) aligns with similar observations in [1], reinforcing that this confidence metric meaningfully captures reasoning quality.
> >
> > Additionally, the choice of "zero-order" optimization offers better memory stability and resource efficiency compared to standard backpropagation. This advantage is theoretically and empirically supported by [3], which demonstrates that forward-pass gradient estimation can match backpropagation performance while maintaining an inference-only memory footprint. In contrast, standard backpropagation requires caching intermediate activations across all layers, which is computationally expensive and memory-intensive during inference. Therefore, our design choice ensures that LTPO remains a lightweight, practical framework for test-time reasoning enhancement.
> >
> > [1] Kang, Zhewei, et al. "Scalable Best-of-N Selection for Large Language Models via Self-Certainty" AI for Math Workshop at ICML (2025).
> >
> > [2] Fu, Yichao, et al. "Deep Think with Confidence" arXiv preprint arXiv:2508.15260 (2025).
> >
> > [3] Malladi, Sadhika, et al. "Fine-Tuning Language Models with Just Forward Passes" NeurIPS (2023).

---

### Official Review · Reviewer_3uYZ · 2025-11-01

**Soundness:** 3
**Presentation:** 4
**Contribution:** 3
**Rating:** 6
**Confidence:** 4

**Summary:**

This paper proposes LTPO - a method that adds certain latent thought token vectors to the input and at test time tries to find optimal vectors via an intrinsic model confidence based reward. After certain iterations at test time, the optimal though vectors are concatenated with the input tokens to obtain the outputs. The paper shows that LTPO outperforms existing methods like zero shot and soft CoT in most cases on math datasets.

**Strengths:**

1. Adding thought tokens that can be optimized via model's own confidence based rewards is a powerful way to improve reasoning.
2. The methods overcomes the efficiency issue of producing long CoT traces.
3. LTPO method outperforms popular techniques like zero shot and soft CoT on math tasks
4. Paper is well written and easy to follow

**Weaknesses:**

1. One major drawback is lack of sufficient experiments. While experiments are conducted on math datasets, the paper can be strengthened by showing the method works on other kinds of tasks as well
2. Lack of generalization experiments for LTPO: extend to other domains

**Questions:**

1. Are the latent thought tokens initialized to [THOUGHT] for every test instance x, or is the optimal thought vector for one instance used as a starting initialization for the next test instance?
2. Howcome the accuracy remains stable when 1 to 16 thought tokens are used? Any thought? Does that mean the more thought tokens being used do not learn any additional signals?

---

> ### Author Response · Authors · 2025-11-20
> **Thanks for the review - Response to Reviewer 3uYZ (Part I)**
>
> Dear Reviewer 3uYZ,
>
> We are sincerely grateful for your informative feedback, which has helped improve our paper (please see the updated paper and appendix). Please see our point-to-point response below.
>
> > Weakness: One major drawback is lack of sufficient experiments. While experiments are conducted on math datasets, the paper can be strengthened by showing the method works on other kinds of tasks as well / Lack of generalization experiments for LTPO: extend to other domains.
>
> Thank you for your constructive feedback. We agree that demonstrating the efficacy of LTPO beyond mathematical reasoning is crucial for establishing it as a general-purpose reasoning framework.
>
> To address this, we have conducted additional experiments on **Commonsense Reasoning** and **Symbolic Reasoning** domains. Following the experimental protocol of our advanced latent reasoning baseline, SoftCoT [1], we evaluated LTPO on the StrategyQA and Date Understanding benchmarks.
>
> **Table 1:** Performance of LTPO vs. SoftCoT vs. Zero-Shot CoT on Strategy QA (commonsense reasoning) and DU  (symbolic reasoning) benchmarks across 2 models, reported in accuracy (%). The optimal results are in **bold** and the suboptimal ones are in _italic_. "DU" indicates the Date Understanding dataset. The '*' symbol indicates the accuracy is reported by the original authors of the SoftCoT paper.
>
> | **Model**    | **Method**      | **StrategyQA** | **DU**    | **Avg.**  |
> |:-------------|:----------------|---------------:|----------:|----------:|
> | LLaMA-3.1-8B-| Zero-Shot CoT   | _71.49_        | 55.60     | 63.55     |
> | Instruct     | SoftCoT*        | 69.04          | _59.04_   | _64.04_   |
> |              | SoftCoT         | 67.34          | 58.40     | 62.87     |
> |              | **LTPO (Ours)** | **72.18**      | **62.00** | **67.09** |
> | Qwen-2.5-7B- | Zero-Shot CoT   | _72.63_        | 64.40     | _68.52_   |
> | Instruct     | SoftCoT*        | 60.61          | 67.52     | 64.07     |
> |              | SoftCoT         | 62.75          | _68.40_   | 65.58     |
> |              | **LTPO (Ours)** | **73.09**      | **76.40** | **74.75** |
>
> The new results demonstrate that LTPO generalizes effectively to non-mathematical domains: LTPO achieves the highest accuracy across all metrics. Notably, on Date Understanding with Qwen-2.5-7B-Instruct, LTPO achieves **76.40%**, outperforming Zero-Shot CoT (**+12.0%**) and SoftCoT (**+8.0%**) by significant margins.
>
> **Conclusion:** These experiments confirm that LTPO is not limited to mathematical problems but is a robust, domain-agnostic framework for enhancing reasoning. We have included these results in the **Section 4.6** and **Appendix C**.
>
> [1] Xu, Yige, et al. "SoftCoT: Soft Chain-of-Thought for Efficient Reasoning with LLMs" ACL (2025).
>
> > Q1: Are the latent thought tokens initialized to [THOUGHT] for every test instance x, or is the optimal thought vector for one instance used as a starting initialization for the next test instance?
>
> Thanks for the question.
>
> We clarify that the latent thought tokens are **initialized from scratch for every test instance**, rather than being carried over from previous instances.
>
> We adopted this instance-specific initialization for two main reasons:
>
> - **Contextual Specificity:** The optimal "thought" vector for one problem (e.g., a geometry problem) is likely distinct from, and potentially detrimental to, the reasoning required for a different problem (e.g., an algebra problem). Resetting the vectors prevents negative transfer between unrelated tasks.
> - **Independence:** Our framework, LTPO, focuses on test-time optimization where each instance is treated as an independent optimization problem. This ensures that the reasoning path is optimized solely to maximize the confidence for the current context without bias from previous interactions, and also, without relying on knowledge gained from other instances.

---

> ### Author Response · Authors · 2025-11-20
> **Thanks for the review - Response to Reviewer 3uYZ (Part II)**
>
> > Q2: Howcome the accuracy remains stable when 1 to 16 thought tokens are used? Any thought? Does that mean the more thought tokens being used do not learn any additional signals?
>
> Thank you for this excellent question. It points to one of the key strengths and differentiators of our proposed LTPO framework, which is highlighted in Figure 2 (left) and discussed in Section 4.5.1.
>
> Your observation is correct: LTPO's performance remains remarkably stable and robust as the number of thought tokens increases from 1 to 16. This is not a sign that the additional tokens are learning no new signals, but rather a demonstration of the robustness of our dynamic optimization process.
>
> Here is a more detailed breakdown:
>
> #### 1. Dynamic Optimization vs. Static Projection
>
> The key difference lies in how the thought tokens are used by our method versus the baselines.
>
> - **SoftCoT (Baseline):** This method relies on a **static linear projection mechanism** that is _trained offline_ to map the assistant model's thoughts into a fixed number of latent vectors to feed the main model. As shown in Figure 2, this static module becomes brittle as the number of latent tokens increases at test time, leading to a sharp drop in performance.
> - **LTPO (Ours):** Our method performs **dynamic test-time optimization**. We do not _train_ a static module. Instead, LTPO treats the K latent thought tokens as a dynamic "workspace". For each new problem, our online policy gradient method actively refines _all_ K vectors to find an optimal state that maximizes the intrinsic confidence reward.
>
> The stability of LTPO stems from its nature as a test-time optimization framework rather than a static projection method.
>
> - **Adaptability:** Unlike SoftCoT, which relies on a fixed linear projection learned offline, LTPO dynamically refines the latent vectors for each specific problem instance. Whether the model is provided with 1 token or 16, the gradient ascent process (Equation 10) actively updates these parameters to their optimal state.
> - **Prevention of Degradation:** As noted in our analysis (Figure 2), SoftCoT degrades sharply with more tokens because its static projector struggles to produce high-quality representations with more tokens without introducing noise. In contrast, LTPO's optimization loop effectively "prunes" non-useful directions in the latent space, ensuring that additional tokens do not disrupt the reasoning process even if they are not strictly necessary.
>
> #### 2. Why Stability is a Strength, Not a Weakness
>
> This stability is precisely the desired behavior. It demonstrates that our optimization process is robust and not-overly sensitive to this hyperparameter:
>
> - **Effective Use of "Workspace":** For a given problem, the optimization will find the best way to use the allotted tokens. If a simple problem only requires the equivalent of "two thoughts" to reach a high-confidence answer, our LTPO process will effectively optimize two tokens and the other 14 will be refined to a state that **does not harm performance**.
> - **No "Curse of Dimensionality":** The failure of SoftCoT shows that for static methods, adding more tokens introduces noise or "confuses" the model, as the static projection fails. The stability of LTPO shows our dynamic optimization **does not suffer from this failure mode**. It can effectively manage the latent "workspace", whether it's small or large, and refine the entire set of vectors toward an optimal configuration.
>
> In summary, the stability is not a sign of redundancy. It is a critical finding that demonstrates LTPO's robustness. Unlike static methods that break, our on-the-fly, test-time optimization can effectively utilize the latent thought tokens it is given, regardless of their quantity, to find a high-confidence reasoning path.
>
> We have included this analysis in **Appendix E**.

---

### Official Review · Reviewer_gpmg · 2025-11-03

**Soundness:** 3
**Presentation:** 3
**Contribution:** 3
**Rating:** 6
**Confidence:** 3

**Summary:**

This paper proposes LTPO, a parameter-free, test-time method that improves LLM reasoning by optimizing latent thought token embeddings with an online policy-gradient loop, guided by a confidence-based intrinsic reward computed from the model’s own output distributions. The method perturbs latent thought vectors with a Gaussian policy, scores them via a top-k log-probability reward, and ascends the estimated gradient; the best latent state found is then used to decode the final answer. Experiments across GSM8K, MATH-500, ASDiv-Aug, and AIME24/25 show consistent gains.

**Strengths:**

1. The paper is generally well-written and structured, followed by the motivation and explanation.
2. LTPO formalizes latent-token search as test-time RL with a simple Gaussian exploration policy and confidence reward.
3. On AIME24/25 benchmarks, LTPO avoids the collapse seen in trained latent methods and surpasses Zero-Shot CoT on several models.

**Weaknesses:**

1. In Figure 2, the best performance is achieved with 1 or 2 thoughts, and increasing the number of thought tokens does not necessarily lead to better results. Since the base model, Llama 3.1 8B, is not capable of generating long chains of thought, I suggest evaluating the method using models with stronger reasoning ability, such as the DeepSeek Distill series.
2. The reward is computed only at latent token positions, but it would be helpful for the authors to justify why this design choice can predict the correctness of the final answer.
3. In Line 190, the authors mention that the reward is “not a perfect proxy for correctness” and refer to Appendix B for discussion. I suggest adding a hybrid reward function that combines model confidence with a verifier to reduce overconfidence bias and better align the reward with true correctness.

**Questions:**

What is the rollout setting during evaluation? Do the authors generate the response only once per question, or perform multiple rollouts? I suggest reporting results using pass@k.

---

> ### Author Response · Authors · 2025-11-20
> **Thanks for the review - Response to Reviewer gpmg (Part I)**
>
> Dear Reviewer gpmg,
>
> We greatly appreciate your insightful and constructive comments, which have helped us improve our paper. Below please see our response to your concerns. In case you find anything unclear, please kindly let us know.
>
> > W1: In Figure 2, the best performance is achieved with 1 or 2 thoughts, and increasing the number of thought tokens does not necessarily lead to better results. Since the base model, Llama 3.1 8B, is not capable of generating long chains of thought, I suggest evaluating the method using models with stronger reasoning ability, such as the DeepSeek Distill series.
>
> Thank you for the insightful observation regarding Figure 2 and the relationship between latent thought tokens and the base model's reasoning capabilities. We agree that evaluating our method on models with stronger reasoning abilities is crucial to validate the scalability and robustness of LTPO.
>
> To address this and evaluate LTPO on a model with state-of-the-art reasoning capabilities, we followed your suggestion to move to a stronger class of models. However, rather than using the suggested DeepSeek Distill series, we selected the **Qwen3-14B** model (with max output tokens set to 64,000) for our new experiments (see added **Section 4.7**).
>
> We chose Qwen3-14B for two strategic reasons that directly address your concern about "stronger reasoning ability":
>
> 1. **Native "Thinking Mode":** Unlike distilled models that mimic reasoning patterns via SFT, Qwen3 features a native "Thinking Mode" engineered with a dynamic thinking budget. This makes it an ideal testbed for observing whether LTPO can effectively steer an intrinsic reasoning process rather than just triggering memorized patterns.
> 2. **Extended Generation Budget:** Qwen3 supports massive context windows, allowing us to set the **max output tokens to 64,000**. This removes the "short chain" constraint completely, allowing the model to generate extensive reasoning trajectories.
>
> **New Results (Section 4.7, Table 5)**: Under this high-compute regime (64k tokens), the Qwen-3-14B baseline is exceptionally strong, achieving **75.00%** average accuracy on the AIME 2024/2025 benchmarks (comparable to OpenAI o1-preview). Crucially, LTPO does **not** saturate or diminish in value; it further boosts this elite baseline to **80.00%** (+5.0% absolute gain on average). To facilitate a direct comparison, we also present the table below:
>
> **Table:** Performance of LTPO vs. Zero-Shot CoT on AIME2024 and AIME2025 with Qwen-3-14B, reported in accuracy (%). The maximum output tokens is set to 64,000. The optimal results are in bold.
>
> | **Method**      | **AIME24** | **AIME25** | **Avg.**  |
> |:----------------|-------------:|-------------:|----------:|
> | Zero-Shot CoT   | 80.00        | 70.00        | 75.00     |
> | **LTPO (Ours)** | **83.33**    | **76.67**    | **80.00** |
>
> This result is significant because it demonstrates that LTPO is not merely a "crutch" for weak models (like Llama-3.1-8B) but a **scalable trajectory optimizer**. Even when the model has a 64,000-token budget to "think" explicitly, optimizing the initial latent thought vectors provides a better starting point for the reasoning trajectory.
>
> We believe these new results on Qwen3-14B answer your question: when applied to a model with strong reasoning ability, LTPO acts as a powerful force multiplier rather than hitting a ceiling.
>
> > W2: The reward is computed only at latent token positions, but it would be helpful for the authors to justify why this design choice can predict the correctness of the final answer.
>
> Thank you for this insightful question.
>
> Although the reward is computed only at the latent token positions, but it is not computed in a vacuum. Due to the transformer's **causal self-attention mechanism**, the output logits at any latent thought token position are conditioned on the entire preceding context. This context includes the full problem statement as well as all prior latent thought tokens.
>
> Therefore, the confidence score (Equation 4) measures how certain the model is about its next prediction, given the full problem and the reasoning path so far. A high reward indicates that the current latent thought vectors have successfully encoded the problem and a "good" reasoning step, leading the model to a confident, peaked output distribution.
>
> Our extensive empirical results validate this hypothesis. The fact that LTPO achieves substantial gains, especially on highly challenging AIME benchmarks where other latent reasoning methods completely fail, demonstrates that this intrinsic confidence-based reward is a powerful and effective proxy for guiding the model toward a correct final answer (even though it is not perfect as discussed in **Appendix B**).

---

> > ### Author Response · Authors · 2025-11-20
> > **Thanks for the review - Response to Reviewer gpmg (Part II)**
> >
> > > W3: In Line 190, the authors mention that the reward is “not a perfect proxy for correctness” and refer to Appendix B for discussion. I suggest adding a hybrid reward function that combines model confidence with a verifier to reduce overconfidence bias and better align the reward with true correctness.
> >
> > Thank you for this constructive suggestion.
> >
> > The core challenge in implementing a verifier-based reward stems from the fundamental design of LTPO as a **test-time-only framework**. At test time, a ground-truth verifier is, by definition, unavailable.
> >
> > A plausible alternative, as you suggest, might be an "LLM-as-a-judge". However, this approach would unfortunately undermine a key strength and contribution of our method: **inference efficiency**. A hybrid reward using an LLM verifier would require generating a full-text solution and then performing a second, separate LLM call to evaluate it... **for every single optimization step**. This overhead would eliminate the significant efficiency gains we demonstrate (as shown in **Table 3** in **Section 4.4**).
> >
> > Therefore, our work deliberately explores the trade-off: we accept the limitations of a confidence-based proxy in exchange for a highly efficient, general, and self-contained test-time enhancement. The strong results, especially the dramatic gains on AIME benchmarks where other methods fail, demonstrate the practical power of this approach.
> >
> > We strongly agree that developing more robust reward signals is a valuable research direction. As we state in Appendix B, "Future work could explore incorporating uncertainty estimation or other signals to create more robust rewards that better align with logical correctness". We believe our work provides a strong foundation for such future explorations.

---

> ### Author Response · Authors · 2025-11-20
> **Thanks for the review - Response to Reviewer gpmg (Part III)**
>
> > Q: What is the rollout setting during evaluation? Do the authors generate the response only once per question, or perform multiple rollouts? I suggest reporting results using pass@k.
>
> Thank you for this question, which allows us to clarify our evaluation protocol and the distinction between our optimization method and stochastic search techniques.
>
> **1. Clarification on the Rollout Setting**
>
> To answer your specific question: **We generate the response only once per question.** The evaluation setting for all reported results in the paper is strictly **single-rollout greedy decoding**.
>
> We suspect the confusion regarding "rollouts" may stem from our use of $T$ optimization steps (e.g., $T=20$) described in Section 3.2. We would like to clarify that these are latent optimization steps, not generative rollouts:
>
> - **Internal Optimization ($T$ steps):** These steps occur entirely within the continuous latent space. They involve forward passes to compute confidence rewards and gradient updates to the latent vectors $H$, but they **do not** involve autoregressive text generation. This process is computationally lightweight compared to full decoding.
> - **Final Generation (1 rollout):** Once the latent vectors are optimized, we perform exactly one autoregressive generation pass using greedy decoding to produce the final answer.
>
> **2. Rationale for Reporting Greedy Accuracy (Pass@1)**
>
> Regarding the suggestion to report `Pass@k`, we carefully selected **Greedy Accuracy (Pass@1)** as our primary metric to ensure scientific rigor, fair comparison, and alignment with our method's contributions.
>
> - **Alignment with Foundational Standards:** The use of greedy decoding is consistent with the foundational literature on LLM reasoning. Notably, **Chain-of-Thought Prompting [1]** and **Zero-Shot CoT [2]** established greedy decoding as the primary protocol for evaluating a model's intrinsic reasoning capabilities. By adhering to this standard, we isolate the model's ability to reason correctly in a single, deterministic attempt, rather than conflating reasoning capability with the probability of success via stochastic search (e.g., Self-Consistency).
> - **Standardization with Latent Reasoning Baselines:** Furthermore, the specific sub-field of latent reasoning has coalesced around this protocol.
>     - **Coconut [3]** explicitly states, "We use greedy decoding for all experiments".
>     - **LatentSeek [4]**, a primary baseline, states, "For all backbones and both prompts, we use greedy decoding for inference".
>     - **SoftCoT [5]** also evaluates using a single reasoning path with greedy decoding. Reporting `Pass@k` would deviate from this established standard, making our results incomparable to prior art.
> - **Optimization vs. Search:** LTPO is designed as a deterministic optimization framework, not a stochastic search method. Our objective is to refine the model's state so that the single most likely path is correct. `Pass@k` rewards coverage (finding one correct answer among many), whereas our method aims for precision (making the best answer correct). High `Pass@1` scores validate that our optimization successfully concentrates probability mass on the correct reasoning path.
> - **Efficiency:** A key contribution of LTPO is its efficiency relative to methods like LatentSeek. As shown in Table 3 of our paper, LTPO is significantly faster than baselines while achieving higher accuracy. Using `Pass@k` would arbitrarily inflate the inference cost by a factor of $k$, obscuring the efficiency gains that are central to our contribution. We believe demonstrating superior performance with a single attempt is a stronger result than doing so with multiple attempts.
>
> [1] Wei, Jason, et al. "Chain-of-Thought Prompting Elicits Reasoning in Large Language Models" NeurIPS (2022).
>
> [2] Kojima, Takeshi, et al. "Large Language Models are Zero-Shot Reasoners" NeurIPS (2022).
>
> [3] Hao, Shibo, et al. "Training Large Language Models to Reason in a Continuous Latent Space" arXiv preprint arXiv:2412.06769 (2024).
>
> [4] Li, Hengli, et al. "Seek in the Dark: Reasoning via Test-Time Instance-Level Policy Gradient in Latent Space" arXiv preprint arXiv:2505.13308 (2025).
>
> [5] Xu, Yige, et al. "SoftCoT: Soft Chain-of-Thought for Efficient Reasoning with LLMs" ACL (2025).

---

### Author Response · Authors · 2025-11-26
**Summary of Rebuttals and Additional Experiments**

Dear Reviewers,

We sincerely thank you for your constructive and insightful comments. During the discussion period, we have carefully addressed all individual questions and updated the paper and the appendix accordingly (all updates are marked as blue).

We are writing to summarize the major updates and additional experiments conducted in response to your collective feedback, which we believe significantly strengthen the paper’s contribution:

1. **Scalability to Strong Reasoners (Addressing Reviewers gpmg, JXR4)** To verify that LTPO acts as a scalable force multiplier rather than just a boost for smaller models, we evaluated **Qwen3-14B** with a massive **64,000 token output budget** on the challenging AIME 2024/2025 benchmarks.

    - **Result:** Even atop this exceptionally strong baseline (which achieves 75.00% average accuracy, 80.00% on AIME24, 70.00% on AIME25), LTPO further boosts performance to **80.00%** (83.33% on AIME24, 76.67% on AIME25).
    - **Significance:** This confirms LTPO provides a better initialization for reasoning trajectories even in high-compute, long-context regimes.
    - We have included these new results in the **Section 4.7 (Table 5)**.

2. **Comparison with LatentSeek (Addressing Reviewers Xm7C, JXR4)** We conducted a comprehensive comparison with the concurrent work _LatentSeek_ (Li et al., 2025).

    - **Performance:** LTPO achieves superior robustness, particularly on AIME, where LatentSeek performance degrades significantly (often <10%).
    - **Efficiency:** LTPO is over **25x faster** than LatentSeek (e.g., ~31s vs ~870s per problem on AIME24, ~24s vs ~708s per problem on AIME25). This highlights LTPO’s advantage as a practical, lightweight test-time optimization framework that avoids the heavy cost of full-text decoding during the optimization loop.
    - We have included these new results in the **Section 4.2 (Table 1)** and **Section 4.4 (Table 3)**.

3. **Generalization to Non-Math Domains (Addressing Reviewer 3uYZ)** To demonstrate domain agility, we extended our evaluation to **Commonsense Reasoning (StrategyQA)** and **Symbolic Reasoning (Date Understanding)**.

    - **Result:** LTPO consistently outperforms both Zero-Shot CoT and SoftCoT across LLaMA-3.1 and Qwen-2.5 backbones (e.g., +12.0% over Zero-Shot CoT and +8.0% over SoftCoT on Date Understanding with Qwen-2.5-7B-Instruct).
    - We have included these new results in the **Section 4.6** and **Appendix C (Table 8)**.

4. **Efficiency Analysis (Addressing Reviewer JXR4)** We clarified that while LTPO introduces an optimization loop, it is often faster than standard Zero-Shot CoT on complex tasks.

    - **Reason:** LTPO tends to find more concise reasoning paths. On AIME 2024, LTPO reduces the average generated token count from 2886 (Zero-Shot CoT) to 1420, resulting in a net reduction in total inference time despite the added optimization overhead, which is small compared to the total inference time (1.86s vs 31.8s).
    - We have included these new results in the **Appendix D (Table 9)**.

We hope these new results address your concerns. We remain available to answer any further questions during the discussion period.

Best regards,

The Authors of Submission 7696

---

### Meta-Review · Area_Chair_zDWD · 2026-01-07

**Summary:**

This paper introduces LTPO, a test-time optimization method that improves LLM reasoning by directly optimizing latent thought token embeddings via a lightweight policy-gradient loop driven by an intrinsic confidence-based reward. Reviewers broadly agree that the idea is well-motivated, technically sound, and clearly presented, with consistent gains across multiple math benchmarks and strong performance on challenging AIME tasks. Key concerns initially centered on evaluation scope (stronger reasoners, non-math domains), missing comparisons to closely related contemporaneous work (LatentSeek), and justification of design choices (top-k confidence reward, policy gradient vs. differentiable alternatives). The rebuttal substantively strengthened the paper by adding experiments on Qwen3-14B with long contexts, direct and convincing comparisons to LatentSeek, non-math generalization results, and a clearer efficiency analysis. Overall, the work now presents a compelling and practical test-time reasoning improvement, though some methodological choices remain heuristic and would benefit from deeper theoretical grounding.

**Reviewer Concerns:**

Concerns addressed -

* Scalability to strong reasoners
* Missing comparison to LatentSeek
* Generalization beyond math
* Efficiency vs. Zero-Shot CoT
* Evaluation clarity

Concerns not addressed -

* Design justification
* Hyperparameter sensitivity

**Reviewer Scores:**

I dont think so.

---

### Decision · Program_Chairs · 2026-01-26

Accept (Poster)